# Marginal Girsanov Reweighting: Stable Variance Reduction via Neural Ratio Estimation

## Abstract

Recovering unbiased properties from biased or perturbed simulations is a central challenge in rare-event sampling. Classical Girsanov Reweighting (GR) offers a principled solution by yielding exact pathwise probability ratios between perturbed and reference processes. However, the variance of GR weights grows rapidly with time, rendering it impractical for long-horizon reweighting. We introduce Marginal Girsanov Reweighting (MGR), which mitigates variance explosion by marginalizing over intermediate paths, producing stable and scalable weights for long-timescale dynamics. Experiments demonstrate that MGR (i) accurately recovers kinetic properties from umbrella-sampling trajectories in molecular dynamics, and (ii) enables efficient Bayesian parameter inference for stochastic differential equations with temporally sparse observations.

## 1 Introduction

Estimating expectations under complex stochastic systems is central to modern machine learning in the sciences and engineering application. Direct sampling from the target law is often infeasible, so reweighting techniques express these expectations as averages under a more tractable reference law. Indeed, these reweighting strategies are fundamental tools in computational chemistry (Mey et al., 2014), mathematics (Beskos & Roberts, 2005), finance (Pascucci, 2011), and machine learning (Domke & Sheldon, 2018). Unfortunately, their practical use is often limited by estimator variance.

This reweighting principle is important in the analysis of stochastic dynamics, where probability laws are defined over path space (Sørensen, 2004; Donati et al., 2017). For example, in molecular dynamics (MD), perturbed potentials are used to overcome free energy barriers due to slow mixing or interest in rare events, and reweighting is then required to recover unbiased quantities such as free energies and transition rates (Torrie & Valleau, 1977; Kästner, 2011; Mey et al., 2014). Similarly, in parameter inference for stochastic differential equations (SDEs), simulations are often carried out under a reference parameter, with likelihood ratios reweighting alternative candidates to form posterior distributions (Li et al., 2020; Ghosh et al., 2022). Across both settings, a common way to compute the importance weights is provided by Girsanov's theorem (Girsanov, 1960).

Girsanov reweighting (GR) (Girsanov, 1960) provides a pathwise ratio, which enables a change of measure between SDEs that share the diffusion but differ in drift (Sørensen, 2004; Donati et al., 2017; 2022). Despite this elegance, two practical limitations arise. First, weight variance grows rapidly with time horizon, causing the exponentiated weights to explode or vanish in long timescales. Second, the weights are inherently tied to specific trajectories. Yet many quantities of interest—such as MD transition probabilities or SDE likelihoods—depend only on endpoints and require marginal ratios that integrate over all intermediate paths. Together, these issues severely restrict the use of Girsanov reweighting in long timescales and large systems.

To address these limitations, we propose a machine learning–based approach: **Marginal Girsanov Reweighting (MGR)**. Instead of relying on full-path Girsanov weights, which become numerically unstable for long trajectories, MGR learns marginal density ratios between end-points of trajectories. The key idea is to leverage accurate short-lag Girsanov weights and iteratively compose them into longer-lag ratios using neural classifiers. We formulate ratio estimation as a binary classification problem, where a neural network distinguishes between samples from reference and target distributions (Menon & Ong, 2016; Choi et al., 2021). By combining the mathematical foundation of

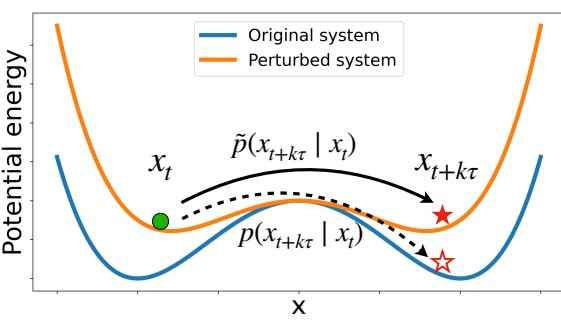 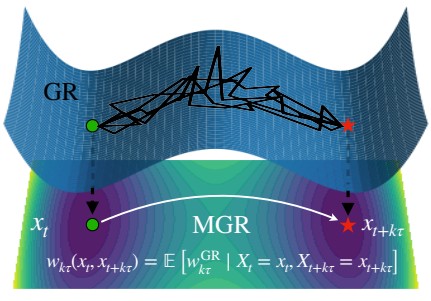

Figure 1: Marginal Girsanov Reweighting (MGR). For the given pairs, MGR defines the marginal weight as expectation of pathwise Girsanov reweighting (GR) factors as introduced in Section 4.1.

Girsanov reweighting with the flexibility of neural ratio estimation, MGR enables reliable inference across domains where traditional estimators break down. To summarize:

- We propose Marginal Girsanov Reweighting (MGR), an iterative learning approach, which estimates transition-based density ratios by marginalizing over intermediate paths.

- We implement MGR as a binary classification task with a weighted cross-entropy objective.

- We demonstrate the effectiveness of MGR on reweighting problems in two distinct domains (i) recovering unbiased thermodynamic and kinetic properties from biased MD simulations, and (ii) parameter posterior inference for SDEs from noisy observations.

## 2 RELATED WORK

**Accelerated sampling via perturbations** In many physical and molecular systems, direct sampling from the target dynamics is computationally infeasible due to slow mixing or the rarity of the events of interest. This has motivated a broad class of perturbation-based accelerated sampling methods that modify the dynamics to facilitate exploration. In molecular dynamics, metadynamics (Huber et al., 1994; Barducci et al., 2008) and umbrella sampling (Torrie & Valleau, 1977; Kästner, 2011) add additional energy (Grubmüller, 1995) to reaction coordinates in order to accelerate exploration. (Swendsen & Wang, 1986; Sugita & Okamoto, 1999; Wang et al., 2011) change the temperature of the system to help cross barriers.

**Reweighting of Dynamics and Path Sampling** Reweighting is a necessary technique to recover the original dynamic from the accelerated data (Kamenik et al., 2022). For thermodynamic quantities, methods such as the weighted histogram analysis method (WHAM) (Gallicchio et al., 2005; Souaille & Roux, 2001) and the multistate Bennett acceptance ratio (MBAR) (Shirts & Chodera, 2008), provide efficient estimators of equilibrium energies and have been applied to multiple windows enhanced sampling. Recent work also (Dibak et al., 2022; Wang et al., 2022; Moqvist et al., 2025; Invernizzi et al., 2022) uses machine learning to estimate the energy differences between different temperatures.

For kinetics, finite-lag transition densities help characterize long time behavior, where intermediate states can be ignored (Wu et al., 2017; Schreiner et al., 2023; Klein et al., 2023; Diez et al., 2024; 2025). With enhanced-simulation data, unbiased kinetics can be recovered by: combining multiple equilibrium ensembles (Mey et al., 2014) within a reversible Markov-state-model framework (Prinz et al., 2011; Husic & Pande, 2018), or directly learning the eigenfunctions of transfer operators from the biased simulations (Devergne et al., 2024). Recently, Girsanov-based path reweighting, which explicitly calculates the Radon–Nikodym derivative in configuration space (Donati et al., 2017; 2022; Schafer & Keller, 2024), has drawn our attention.

**Bayesian inference for SDEs** In classical Bayesian inference for parametric SDEs, the drift and diffusion structures are given analytically, but contain unknown parameters. With only discrete and sparsely observed data, the likelihood of these observations is analytically intractable. A common strategy is to simulate latent paths under a reference SDE that is numerically stable or computationally convenient, and then express the likelihood ratio relative to the target SDE via Girsanov's theorem (Girsanov, 1960; Sørensen, 2004). This approach underlies a wide range of inference

algorithms, including particle MCMC (Andrieu et al., 2010), and series-expansion–based inference (Lyons et al., 2014; Ghosh et al., 2022). It has wide applications in life sciences (Fuchs, 2013; Bunin, 2017) and financial modeling (Jones, 1998; Eraker, 2001). Recent advances in simulation-based inference (SBI) provide neural posterior by amortizing inference over large collections of simulated data pairs (Cranmer et al., 2020; Gloeckler et al., 2024; Cai et al., 2023). These methods require extensive simulation across a broad parameter range, and aim to learn a surrogate and black-box simulator. In our paper, we focus on the former—classical Bayesian inference—which operates in the physics-informed regime where the underlying SDE mechanism is known. The main challenge arises from the high variance of path-space likelihood ratios.

## 3 PRELIMINARIES

### 3.1 BROWNIAN DYNAMICS AND MIXING

A diffusion process described by a stochastic differential equation (SDE) satisfies

$$\mathrm{d}X_t = f(X_t, t)\mathrm{d}t + g(t)\,\mathrm{d}W_t, \quad X_0 = x_0, \tag{1}$$

where $X_t \in \mathbb{R}^d$ denotes the state of the stochastic process at time $t$, $f(\cdot, t) : \mathbb{R}^d \to \mathbb{R}^d$ is a drift vector, $g(t) \in \mathbb{R}$ is the diffusion coefficient, and $W_t \in \mathbb{R}^d$ is a standard Wiener process. In a fixed time horizon $t \in [0, T]$, the associated path probability measure induced by Eq. 1 is denoted by $\mu$.

However, direct simulation from the target law is often infeasible (Sørensen, 2004; Vanden-Eijnden et al., 2010), such as slow mixing in molecular dynamics and unknown parameters in Bayesian inference, as illustrated in Section 2. This motivates us to introduce a perturbation to the drift, replacing $f(\cdot, t)$ with $\tilde{f}(\cdot, t)$, while keeping the diffusion coefficient $g(t)$ unchanged. The associated path probability measure is denoted by $\tilde{\mu}$.

Although such perturbed dynamics are easier to simulate, they alter the underlying probability measure and thus distort the statistics of the original system. By calculating the Radon-Nikodym derivative $\frac{\mathrm{d}\mu}{\mathrm{d}\tilde{\mu}}(\mathbf{x})$ between two path measures, we can analyze the original dynamics in Eq. 1 by reweighting the trajectories $\mathbf{x} = \{x_t\}_{t \in [0,T]}$ under the perturbed paths.

### 3.2 GIRSANOV REWEIGHTING THEORY

A common approach to compute the Radon–Nikodym derivative of $\mu$ respect to $\tilde{\mu}$ is provided by Girsanov's theorem (Girsanov, 1960; Donati et al., 2017). If two diffusion processes share the same diffusion coefficient but have different drifts, their path measures can be transformed from one to the other.

Here, we consider the trajectory segment under the perturbed dynamics, $\mathbf{x}_{t,\tau} = \{x_s\}_{s=t}^{t+\tau}$ from $x_t$ to $x_{t+\tau}$, $t \in [0, T-\tau]$. Using an Euler–Maruyama discretization, the corresponding discrete-time trajectory $\{x_t = x^0, x^1, x^2, \ldots, x_{t+\tau} = x^N\}$ is observed with discretization step $\Delta t = \tau/N$. Then the likelihood ratio between the original path $\mu$ and the perturbed path $\tilde{\mu}$, conditional on the same starting state $x_t$, can be calculated as

$$\log w_\tau^{\mathrm{GR}}(\mathbf{x}_{t,\tau}) = \log \frac{\mathrm{d}\mu}{\mathrm{d}\tilde{\mu}}(\mathbf{x}_{t,\tau}|x_t)$$

$$\approx \sum_{k=0}^{N-1} \left( \frac{\left(f(x^k, t^k) - \tilde{f}(x^k, t^k)\right)^\top}{g(t^k)} \sqrt{\Delta t}\xi^k - \frac{\Delta t}{2} \left\| \frac{\left(f(x^k, t^k) - \tilde{f}(x^k, t^k)\right)}{g(t^k)} \right\|^2 \right), \tag{2}$$

where $\sqrt{\Delta t}\,\xi^k = \dfrac{x^{k+1} - x^k - \tilde{f}(x^k, t^k)\,\Delta t}{g(t^k)} \overset{\mathrm{i.i.d.}}{\sim} \mathcal{N}(0, \Delta t \cdot I_d)$ represents the Wiener increment associated with the simulation step $x^k \to x^{k+1}$ under the perturbed dynamics.

Eq. 2 provides a solution for recovering original properties from perturbed simulations. However, two major difficulties arise in practice. First, in computing $\log w_\tau^{\mathrm{GR}}$, long time horizons $\tau$ will introduce an accumulation of noise terms, e.g. $\xi^k$ in Eq. 2, which causes the variance to grow with trajectory length. Upon exponentiation, the weights will explode or vanish. Detailed analysis can be found in Appendix A. Second, when comparing the $\tau$-lag transition probabilities from $x_t$ to $x_{t+\tau}$

under the perturbed and original dynamics, Girsanov reweighting yields a ratio along a specific simulated path, which is a biased proxy for the true marginal quantity (see Section 4.1 for details). Motivated by these two problems, our work develops a model that can estimate a more stable and accurate marginal weights for long time intervals and large systems.

# 4 MARGINAL GIRSANOV REWEIGHTING

## 4.1 SETUP

In stochastic dynamics, probability laws are typically defined over path space, capturing either a single long trajectory or mutiple trajectories $\mathbf{x} = \{x_t\}_{t \in [0,T]}$ under a given stochastic process. However, in many practical applications, our focus shifts from full paths to the paired data $\{(x_t, x_{t+\tau})\}_{t \in [0, T-\tau]}$ between two time points separated by a fixed lag time $\tau$. Under the standard ergodic assumption, the transition-based representation provides meaningful dynamical information.

For example, in molecular dynamics (MD), the analysis of kinetics and transport properties (Prinz et al., 2011; Tiwary et al., 2015; Mardt et al., 2018) is often carried out. The central object is then the transition probability between metastable states over a specified lag time. Similarly, in SDE parameter inference, discrete-time observations are available at sparse intervals of time horizons, and inference relies on computing the likelihood of these observation pairs (Sørensen, 2004; Golightly & Wilkinson, 2008). In these contexts, the object of interest is not the full path law $\mu(\mathbf{x})$, but rather the finite-time transition probability $p\left(x_{t+\tau} \mid x_t\right)$ induced at two time points.

Let $p\left(x_{t+\tau} \mid x_t\right)$ denote the transition probability induced by the original process in Eq. 1, and let $\tilde{p}\left(x_{t+\tau} \mid x_t\right)$ denote the transition probability under the perturbed dynamics. The ratio between these two transition quantities defines a reweighting factor $w_\tau(x_t, x_{t+\tau}) = \frac{p(x_{t+\tau}|x_t)}{\tilde{p}(x_{t+\tau}|x_t)}$.

Instead of computing the intractable conditional density directly, we consider the joint distribution over such pairs. Letting $\rho_\tau(x_t, x_{t+\tau})$ and $\tilde{\rho}_\tau(x_t, x_{t+\tau})$ denote the joint densities under the original and perturbed processes respectively, the reweighting ratio can be expressed as

$$w_\tau(x_t, x_{t+\tau}) = \frac{p\left(x_{t+\tau} \mid x_t\right) \rho\left(x_t\right)}{\tilde{p}\left(x_{t+\tau} \mid x_t\right) \rho\left(x_t\right)} = \frac{\rho_\tau\left(x_t, x_{t+\tau}\right)}{\tilde{\rho}_\tau\left(x_t, x_{t+\tau}\right)}.$$

This representation highlights that the ratio of transition density ratio can be seen as the ratio of joint distributions. Furthermore, it admits a natural transformation in terms of the Radon–Nikodym derivative over path space

$$w_\tau(x_t, x_{t+\tau}) = \mathbb{E}_{\tilde{\mu}}\left[\frac{\mathrm{d}\mu}{\mathrm{d}\tilde{\mu}}(\mathbf{x}_{t,\tau}) \mid X_t = x_t, X_{t+\tau} = x_{t+\tau}\right]. \tag{3}$$

Detailed proof can be found in Appendix B. Based on the pathwise Girsanov weights $\frac{\mathrm{d}\mu}{\mathrm{d}\tilde{\mu}}(\mathbf{x}_{t,\tau})$, we propose the Marginal Girsanov Reweighting (MGR) to learn the reweighting factor for given paired data with long time interval (see Figure 1).

## 4.2 TRAINING ALGORITHM

We denote by $w_{k\tau}(x_t, x_{t+k\tau})$ the likelihood ratio from configuration $x_t$ at time $t$ to $x_{t+k\tau}$ at time $t + k\tau$. For the case $k = 1$, i.e., over a short lag time, Girsanov reweighting $w_\tau^{\text{GR}}$ in Eq. 2 provides a relatively stable and pathwise estimate of the ratio. Our goal is to develop a Marginal Girsanov Reweighting (MGR) approach that can reliably estimate the ratio for longer lag times with $k \gg 1$.

Our method, MGR, adopts an iterative training strategy based on either a long discretized simulation trajectory or multiple discretized trajec-

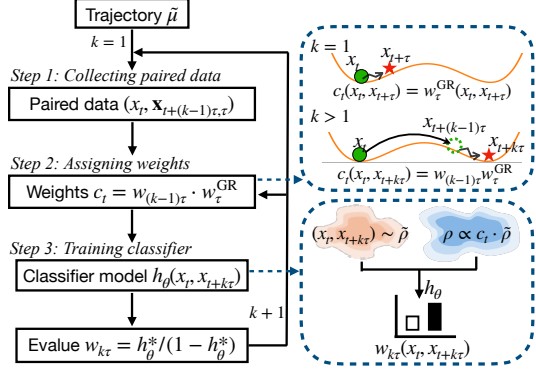

Figure 2: The training algorithm of MGR as illustrated in Sections 4.2 and 4.3

tories under perturbed path $\tilde{\mu}$. Suppose the ratio $w_{(k-1)\tau}$ has already been obtained. Then the ratio $w_{k\tau}$ at lag time $k\tau$ can be constructed as follows:

**Step 1.** We collect pairs of the form $\big(x_t, \mathbf{x}_{t+(k-1)\tau,\tau}\big)$ for $t \in [0, T - k\tau]$ from the perturbed simulation trajectory, where each pair consists of a state and a trajectory segment.

**Step 2.** In this step, we construct approximations of perturbed and original distributions of $(x_t, x_{t+k\tau})$ based on the following identity:

$$\mathbb{E}_{\rho_{k\tau}(x_t, x_{t+k\tau})}[O(x_t, x_{t+k\tau})] = \mathbb{E}_{\rho_{(k-1)\tau}(x_t, x_{t+(k-1)\tau}) \cdot \mu(\mathbf{x}_{t+(k-1)\tau,\tau} | x_{t+(k-1)\tau})}[O(x_t, x_{t+k\tau})]$$
$$= \mathbb{E}_{\tilde{\rho}_{(k-1)\tau}(x_t, x_{t+(k-1)\tau}) \cdot \tilde{\mu}(\mathbf{x}_{t+(k-1)\tau,\tau} | x_{t+(k-1)\tau})}[c_t\, O(x_t, x_{t+k\tau})]\,,$$

for any bounded measurable test function $O(\cdot,\cdot) : \mathbb{R}^d \times \mathbb{R}^d \to \mathbb{R}$. The first equality holds because $\rho_{k\tau}$ can be regarded as the marginal distribution of $(x_t, x_{t+k\tau})$ defined by $\rho_{(k-1)\tau}(x_t, x_{t+(k-1)\tau}) \cdot \mu(\mathbf{x}_{t+(k-1)\tau,\tau} \mid x_{t+(k-1)\tau})$, and the second equality follows from the principle of importance sampling, with weight

$$c_t = w_{(k-1)\tau}(x_t, x_{t+(k-1)\tau}) \cdot w_\tau^{\mathrm{GR}}\big(\mathbf{x}_{t+(k-1)\tau,\tau}\big)\,. \tag{4}$$

Since $w_{(k-1)\tau}$ is inherited from the previous iteration and $w_\tau^{\mathrm{GR}}$ can be computed as described in Section 3.2, the pairs collected in *Step 1* can be used to approximate the perturbed and original joint distributions $\tilde{\rho}_{k\tau}$ and $\rho_{k\tau}$ as

$$\tilde{\rho}_{k\tau}(x, y) \approx \frac{1}{T - k\tau} \sum_t \delta(x - x_t)\, \delta(y - x_{t+k\tau}),$$

$$\rho_{k\tau}(x, y) \approx \sum_t \frac{c_t\, \delta(x - x_t)\, \delta(y - x_{t+k\tau})}{\sum_{t'} c_{t'}},$$

where $\delta$ denotes the Dirac delta function. When accurate estimates of $w_{(k-1)\tau}$ are available and the dataset size is sufficiently large, the above approximations can be shown to be consistent. A detailed proof is provided in Appendix C.

**Step 3.** Finally, using the joint distributions $\tilde{\rho}_{k\tau}(x, y)$, $\rho_{k\tau}(x, y)$ in *Step 2*, we approximate the marginal weight $w_{k\tau}(x, y)$ by a classifier-based density ratio estimator (see Section 4.3 for details). The inferred ratio from the optimal binary classifier is then used for the next iteration.

This three-step procedure effectively extends short-time pathwise Girsanov weights to long-time marginal ratio. First, we train the model $w_\tau$ using the short $\tau$-lag Girsanov weights. Then, we iterate three steps above to progressively learn $w_{k\tau}$ until $k \gg 1$. The workflow is summarized in Figure 2. In the next subsection, we introduce the ratio estimation method used in MGR *Step 3*.

### 4.3 CLASSIFIER-BASED DENSITY RATIO ESTIMATION

Many machine learning methods have been proposed for density ratio estimation (Menon & Ong, 2016; Choi et al., 2021; 2022; Yu et al., 2025). A widely used approach is probabilistic classification (Menon & Ong, 2016), which reformulates likelihood ratio estimation as a binary classification task. In this setting, a binary classifier with a sigmoid output $h(\cdot,\cdot) : \mathbb{R}^d \times \mathbb{R}^d \to [0, 1]$ is trained to discriminate between paired samples $(x_t, x_{t+k\tau})$ drawn from the perturbed distribution $\tilde{\rho}_{k\tau}$ and the original distribution $\rho_{k\tau}$.

We define the optimal classifier $h_\theta^*$ as the probability that a given pair $(x_t, x_{t+k\tau})$ comes from $\rho_{k\tau}$, i.e., $h_\theta^*(x_t, x_{t+k\tau}) = \rho_{k\tau}(x_t, x_{t+k\tau})/\left(\rho_{k\tau}(x_t, x_{t+k\tau}) + \tilde{\rho}_{k\tau}(x_t, x_{t+k\tau})\right)$. Then, the density ratio can be estimated as

$$w_{k\tau}(x_t, x_{t+k\tau}) = \frac{\rho_{k\tau}(x_t, x_{t+k\tau})}{\tilde{\rho}_{k\tau}(x_t, x_{t+k\tau})} = \frac{h_\theta^*(x_t, x_{t+k\tau})}{1 - h_\theta^*(x_t, x_{t+k\tau})}. \tag{5}$$

Unlike standard density ratio estimation, where samples from both distributions are available and the ratio can be learned via cross-entropy loss, our setting in MGR is different. We only have samples from the perturbed distribution $\tilde{\rho}_{k\tau}$ together with the corresponding weight $c_t$ assigned to each sample. To address this, we employ a weighted cross-entropy loss:

$$\mathcal{L}(\theta) = -\mathbb{E}_t \left[ c_t \log h_\theta(x_t, x_{t+k\tau}) + \log\left(1 - h_\theta(x_t, x_{t+k\tau})\right) \right], \tag{6}$$

where $(x_t, x_{t+k\tau})$ denotes the paired data collected from the perturbed trajectory with lagtime $k\tau$. The weights $c_t$ are constructed from the output of the previous model $w_{(k-1)\tau}$ and short-lag Girsanov reweighting $w_\tau^{\mathrm{GR}}$ according to Eq. 4, and are normalized over the entire dataset.

After sufficient training on either a single long trajectory or multiple trajectories, the estimator $w_{k\tau}$ in Eq. 5 converges to the marginal ratio, and then serves as a component for the next iteration. Through the iterative training scheme, MGR can attain stable marginal ratios under long lag times and in complex systems. The complete training procedure is summarized in Algorithm 1 and illustrated in Figure 2. For completeness, we compare a range of ablations and discuss other potential model choices (see Appendix F.1), but defer a more systematic investigation to future work.

## 5 EXPERIMENTS

### 5.1 MOLECULAR DYNAMICS

MGR has particularly useful applications in the analysis of kinetic properties in molecular dynamics (MD) (Donati et al., 2017; 2022). In particular, it can be employed in the construction of Markov state models (MSMs) (Prinz et al., 2011) at a given lag time.

In an MSM framework, the dynamics are characterized by a transition probability matrix $P_\tau$, where each entry is obtained by normalizing the corresponding cross-correlation $C_{ij}(\tau)$. This quantity measures the probability of observing a transition from state $i$ at time $t$ to state $j$ at time $t + \tau$. Under biased or enhanced sampling trajectories $\tilde{\mu}$, we want to recover the unbiased properties using reweighting. For each observed transition pair $(x_t, x_{t+\tau})$, a marginal weight $w_\tau(x_t, x_{t+\tau})$ is assigned to the cross-correlation $C_{ij}(\tau) = \mathbb{E}_{\tilde{\mu}}\left[w_\tau(x_t, x_{t+\tau})\mathbf{1}_{B_i}(x_t)\mathbf{1}_{B_j}(x_{t+\tau})\right]$, where $w_\tau(x_t, x_{t+\tau})$ is estimated by MGR. In our evaluation, we focus on the following several key indicators. Detailed definitions and explanations are provided in Appendix E.1.

- Effective sample size (ESS): Since the true transition density ratio is analytically intractable, relative ESS is then used as a reference indicator for weight stability (Freeman, 1966). A larger ESS implies more reliable statistical estimates, and the results are shown in Appendix E.1.

- Implied timescales (ITS): Each eigenvalue $\lambda_i(\tau)$ of the transition probability matrix defines an implied timescale $t_i(\tau) = -\frac{\tau}{\log \lambda_i(\tau)}$. We examine several dominant implied timescales (ITS) as well as their sum. For each individual ITS, it shows the slow physical timescale of the system. For their sum, an accurate tracking of the exponential decay with increasing lag time is expected, which reflects the intrinsic relaxation behavior of the system dynamics.

- Dominant eigenfunctions and stationary distribution: The leading eigenfunctions characterize the slow dynamical modes, while the stationary distribution represents the long-time equilibrium of the system. Accurate recovery of these quantities is essential for validating kinetic properties.

We primarily compare the results obtained from Marginal Girsanov Reweighting (MGR) with those produced by pathwise Girsanov reweighting (GR) (Donati et al., 2017). Guidelines for selecting a suitable short lag $\tau$ and network for training in MGR are provided in the Appendices F.2 and F.4.

### 5.1.1 ONE DIMENSIONAL FOUR WELL

We first consider a one-dimensional four-well potential system (Prinz et al., 2011), which serves as a prototypical example for testing reweighting methods. The unbiased energy landscape contains four metastable states separated by barriers, with the two intermediate wells located at higher energies. To accelerate sampling, we introduce a biased potential that lowers the energy of the two intermediate wells. Detailed energy function and simulation information can be found in Appendix E.1.1.

Figure 3 presents the results for the four-well system. The left panel shows the dominant left eigenfunctions at lagtime $300\Delta t$ ($\Delta t = 0.001$). GR provides partial correction but still exhibits clear deviations, particularly in the central wells. By contrast, MGR yields eigenfunctions that are in close agreement with the unbiased results. It demonstrates that MGR enables accurate recovery of both equilibrium and kinetic properties. The results of dominant right eigenfunctions and the ESS

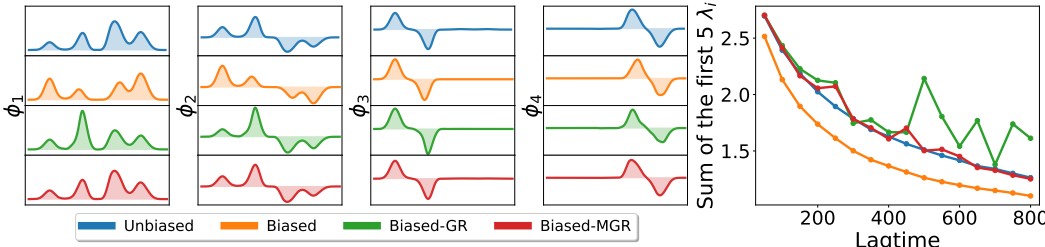

Figure 3: Results for the one-dimensional four-well potential system. **Left:** Dominant left ($\{\phi_i\}$) eigenfunctions of the transition matrix at lagtime $300\Delta t$. ($\phi_1$ denotes the stationary distribution.) Unbiased results serve as reference, while Biased trajectories, Biased reweighted by GR (Biased-GR), and Biased reweighted by MGR (Biased-MGR) are compared. **Right:** sum of the first 5 eigenvalues, reflecting the intrinsic relaxation behavior of the dynamics, as a function of lag time.

curve are shown in Figure S.2, where the ESS for GR decays rapidly at long lag times, reflecting the instability. In contrast, our model MGR maintains substantially higher ESS across lag times, stabilizing reweighting .

We also compare the sum of the first five eigenvalues of the transition matrix in the right panel. The results of MGR closely follow the unbiased reference and exhibit smoother exponential decay, whereas GR suffers from strong fluctuations. This indicates that MGR more accurately captures the intrinsic relaxation dynamics of the system. For completeness, the several individual dominant implied timescales (ITS) are reported in Figure S.3.

### 5.1.2 ALANINE DIPEPTIDE

We next consider alanine dipeptide (Donati et al., 2017; Wu et al., 2020), a widely used benchmark system in molecular dynamics for studying conformational transitions. The backbone torsion angles $\phi$ and $\psi$ are two important reaction coordinates, containing several important metastable basins. Sampling transitions between these basins is challenging due to the presence of high free-energy barriers. To accelerate exploration, we perform biased simulations by introducing umbrella potentials along $\phi$ and $\psi$, which distort the original equilibrium distribution and transition probabilities. Detailed energy function and simulation information can be found in Appendix E.1.2.

Figure 4 shows the results for alanine dipeptide. a) reports the sum of the first five eigenvalues as a function of lag time. MGR consistently follows the unbiased reference, whereas GR shows clear

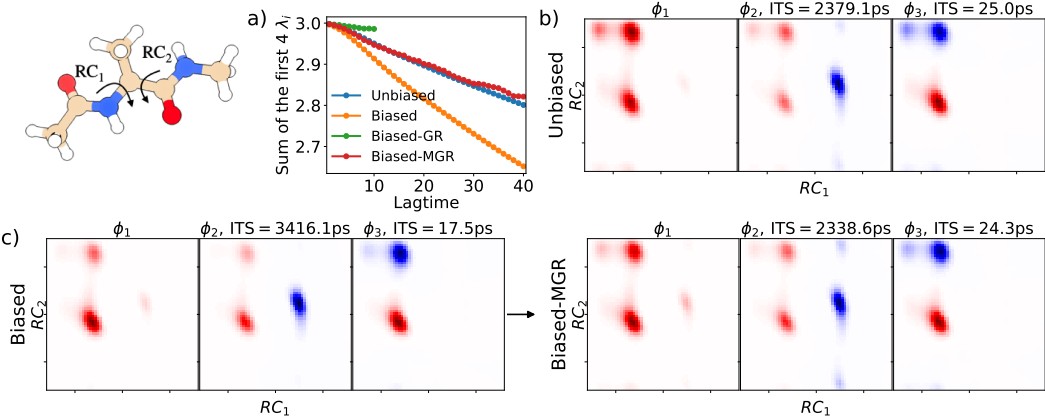

Figure 4: Results for the 22 atoms Alanine Dipeptide system. **a)** sum of the first five eigenvalues, reflecting the intrinsic relaxation behavior of the dynamics, as a function of lag time (lagtime unit is 40fs). **b)** Dominant left eigenfunctions of the transition matrix with corresponding ITS values are shown in the title at lag time 1.6ps. Unbiased results serve as reference. **c)** Dominant left eigenfunctions from the biased trajectory and the MGR-recovered results. GR fails to construct a convergent Markov state model due to excessive variance.

deviations. Due to the large variance and low ESS values, GR fails to provide a reliable MSM for lag times exceeding $10\tau = 400$fs. b) shows the dominant left eigenfunctions and the corresponding implied timescales (ITS) at lag time $40\tau = 1.6$ps under the unbiased trajectory. These serve as our evaluation reference. Using MGR, we reweight the biased trajectory to recover the unbiased properties. Since GR has already failed at this timescale, only MGR results are reported in c). Both the eigenfunctions and implied timescales demonstrate that MGR successfully reproduces the unbiased behavior. Detailed comparisons of ESS and individual ITS are provided in Figure S.6.

These two examples demonstrate that MGR achieves consistently strong performance in molecular dynamics. It maintains stable effective sample sizes, produces reliable implied timescales, and accurately recovers equilibrium and kinetic properties, even under large system and at long lag times where traditional GR fails. These results highlight the effectiveness of MGR, especially in mitigating variance growth and removing path dependence. As a result, MGR provides a practical framework for reweighting biased simulations in MD.

## 5.2 BAYESIAN INFERENCE OF SDE PARAMETERS

Another important application of MGR lies in the Bayesian posterior inference of SDEs (Sørensen, 2004; Golightly & Wilkinson, 2008; Li et al., 2020; Ghosh et al., 2022). MGR is used to estimate the parameters of a stochastic differential equation from discrete observations. Detailed illustration can be found in Appendix E.2.

Consider a $d$-dimensional diffusion process, whose drift term contains unknown parameters $\theta$. Instead of simulating the SDE for every candidate $\theta$, we set a reference parameter $\theta_0$, and generate reference trajectories under $\theta_0$.

Through the MGR model introduced in Section 4, we utilize short-lag Girsanov weights $w_\tau^{\text{GR},\theta}(\mathbf{x}_{t,\tau}) = \frac{\mathrm{d}\mu_\theta}{\mathrm{d}\tilde{\mu}_{\theta_0}}(\mathbf{x}_{t,\tau})$ in Eq. 2 iteratively, where the drift difference is $f(\cdot, t; \theta) - f(\cdot, t; \theta_0)$. After training, stable marginal weights $w_{k\tau}^\theta(\cdot, \cdot)$ are used to estimate the likelihood of sparse observations $\mathbf{y} = \{y(t=0) = y^0, y^1, ..., y(t=T) = y^K\}$, where $k\tau = T/K$:

$$w_{k\tau}^\theta(\mathbf{y}) = \frac{p(\mathbf{y} \mid \theta)}{p(\mathbf{y} \mid \theta_0)} = \Pi_i \frac{p(y^{i+1} \mid y^i; \theta)}{p(y^{i+1} \mid y^i; \theta_0)} = \Pi_i w_{k\tau}^\theta(y^i, y^{i+1}).$$

This allows us to construct a stable likelihood ratio based on the given paired observations $\{(y^i, y^{i+1})\}$. Consequently, posterior expectations can be evaluated by importance sampling with observations $\mathbf{y}$:

$$\mathbb{E}[\theta \mid \mathbf{y}] = \int \theta \, p(\theta \mid \mathbf{y}) \mathrm{d}\theta \approx \frac{\sum_\theta \theta \, w_{k\tau}^\theta(\mathbf{y}) p(\theta)}{\sum_\theta w_{k\tau}^\theta(\mathbf{y}) p(\theta)}.$$

Leveraging short-lag, pathwise Girsanov factors, MGR constructs reliable likelihood ratios over long observation intervals and time horizons, which enables accurate posterior estimation of the parameters. In this example, we primarily compare MGR with the particle marginal Metropolis–Hastings (PMMH) algorithm (Golightly & Wilkinson, 2008; Hoffman et al., 2014) and a variational inference (VI) approach (Ghosh et al., 2022).

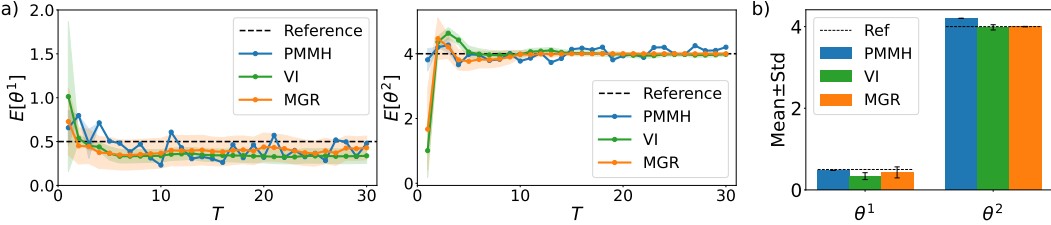

Figure 5: Results for Bayesian inference in the Graph Ornstein–Uhlenbeck process. **a)** Posterior estimates of $\theta^1$ and $\theta^2$ under different observation horizons. Estimation of $\theta^1$ is challenging, and MGR yields more stable posteriors. For $\theta^2$, MGR achieves highly concentrated and precise estimates. **b)** Posterior estimates with mean and standard deviation at the longest observation horizon $T = 30$.

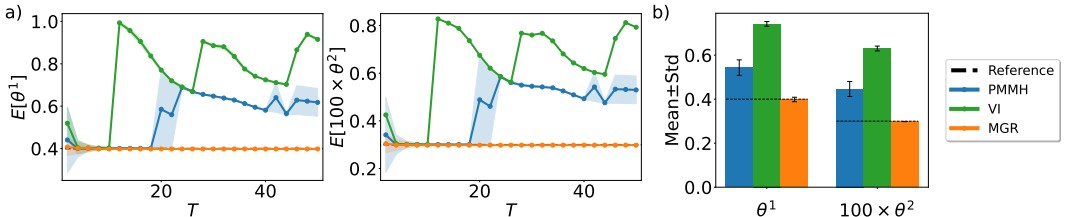

Figure 6: Results for Bayesian inference in the Stochastic Lotka-Volterra system. **a)** Posterior estimates of $\theta^1$ and $100 \times \theta^2$ under different observation horizons. **b)** Posterior distributions by the mean of all time horizons.

### 5.2.1 GRAPH ORNSTEIN-UHLENBECK

we first consider the Ornstein–Uhlenbeck (OU) process (Courgeau & Veraart, 2022) defined on a graph structure,

$$\mathrm{d}X_t = -A(\theta^1)(X_t - B(\theta^2))\,\mathrm{d}t + \sigma\,\mathrm{d}W_t, X_t \in \mathbb{R}^2,\ A(\theta^1) = \begin{pmatrix} 1 & -\theta^1 \\ -\theta^1 & 1 \end{pmatrix},\ B(\theta^2) = \begin{pmatrix} \theta^2 \\ 1 \end{pmatrix},$$

where $\sigma = 0.5$, and $t \in [0, 30]$. We set the reference parameter as $\theta_0^1 = 0.5, \theta_0^2 = 4$, and a latent trajectory are generated under this reference parameter. We consider a grid of 225 candidate parameters, where the detailed simulation settings and parameters candidates are provided in Appendix E.2.1. For the observations, we sample every unit time step $t = 1$ along the simulation path. At each sampling point, Gaussian noise is introduced to obtain observations $\mathbf{y} = \{y(t=0) = y^0, y^1, \ldots, y(t=30) = y^{30}\}, y_i \sim \mathcal{N}(x_i, 0.25I)$.

Figure 5 reports the posterior estimates under different observation trajectory lengths. We observe that $\theta^1$ is more difficult to identify: VI exhibits noticeable bias, while PMMH shows substantial fluctuations and unstable convergence. In contrast, MGR consistently yields better estimates across different horizons. For $\theta^2$, which is easier to infer due to its stronger signal in the dynamics, MGR again provides highly accurate and concentrated posteriors, clearly outperforming the alternatives.

### 5.2.2 STOCHASTIC LOTKA-VOLTERRA

Finally, we consider the stochastic Lotka–Volterra model (Wilkinson, 2018), which describes a population comprising of two competing species: prey grows intrinsically at rate $\theta^1$ and is depleted due to predators at strength $\theta^2$; predator dies intrinsically at rate $\theta^1$ but increases through predation with the same interaction strength $\theta^2$. This system can be defined by

$$\mathrm{d}X_t = A(X_t, \boldsymbol{\theta})\,\mathrm{d}t + \sigma\,\mathrm{d}W_t, X_t \in \mathbb{R}^2,\ A(X_t, \boldsymbol{\theta}) = \begin{pmatrix} \theta^1 X_t^1 - \theta^2 X_t^1 X_t^2 \\ \theta^2 X_t^1 X_t^2 - \theta^1 X_t^2 \end{pmatrix},$$

where $\sigma = 0.1$ and $t \in [0, 50]$. We set the reference parameter as $\theta_0^1 = 0.4, \theta_0^2 = 0.003$, and simulate a latent trajectory under this reference parameter. We also consider a grid of 225 candidate parameters, where the detailed simulation settings and parameters candidates are provided in Appendix E.2.2. For the observations, we sample every unit time step $t = 2$ along the simulation path. At each sampling point, Gaussian noise is introduced to obtain observations $\mathbf{y} = \{y(t=0) = y^0, y^1, \ldots, y(t=50) = y^{25}\}, y_i \sim \mathcal{N}(x_i, 25I)$.

Figure 6 reports the posterior estimates under different observation trajectory lengths. As the time horizon $T$ (and thus the number of observations) increases, both PMMH and VI exhibit growing bias and variability. In contrast, MGR remains essentially flat across $T$ and closely tracks accurate posterior estimates of the parameters.

By estimating the marginal likelihood ratio between paired observations, MGR can effectively avoid weight degeneracy that afflicts the baseline methods. Across these two examples, MGR shows reliable and efficient Bayesian posterior estimates under both long lag times and extended time horizons, which makes it a practical tool for posterior inference in complex stochastic systems.

## 6 CONCLUSION

Our work introduces Marginal Girsanov Reweighting (MGR), an innovative approach for estimating unbiased properties from perturbed paired data. Based on biased pathwise Girsanov reweighting (GR)—which computes the ratio along a specific trajectory—MGR learns a marginal estimator by integrating over intermediate states. This approach mitigates the variance blow-up of GR and yields an unbiased estimate of the transition probability under long lags.

We show that MGR can be implemented using standard density ratio ratio estimation setups with only minor adaptions, and outperforms several base-lines on benchmarks spanning multiple different fields.

Despite MGR's advantages, challenges remain. When the perturbed drift deviates substantially from the reference dynamics, GR weights become unstable, which in turn hampers MGR training. Careful choice of reaction coordinates is therefore essential in practice. Moreover, while we implemented MGR using standard classifier-based ratio estimation, richer neural estimators and architectures hold promise for further gains. In addition, given the rapid development of physical systems and molecular dynamics, an interesting direction is to extend MGR toward simulation-based Bayesian inference and multi-ensemble estimators. Exploring these directions will broaden the scope of MGR, paving the way for robust, ML-driven reweighting methods applicable across scales, domains, and dynamical systems.

## ACKNOWLEDGEMENTS

This work was partially supported by the Wallenberg AI, Autonomous Systems and Software Program (WASP) funded by the Knut and Alice Wallenberg Foundation and by the National Natural Science Foundation of China (NSFC) under grant number 12171367. Preliminary results were enabled by resources provided by the National Academic Infrastructure for Supercomputing in Sweden (NAISS) at Alvis (project: NAISS 2025/22-463), partially funded by the Swedish Research Council through grant agreement no. 2022-06725.

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

## A  VARIANCE OF GIRSANOV REWEIGHTING

We analyze the variance behavior of Girsanov reweighting over a continuous time interval $[t, t + \tau]$. According to Girsanov theory Girsanov (1960), the log-weight under the Girsanov transformation can be expressed as

$$\log w_\tau^{\mathrm{GR}}(\mathbf{x}_{t,\tau}) = \int_t^{t+\tau} u(x_s, s)^\top \mathrm{d}W_s - \frac{1}{2} \int_t^{t+\tau} \|u(x_s, s)\|^2 \, \mathrm{d}s,$$

where $u(x_t, t) := \frac{f(x_t, t) - \tilde{f}(x_t, t)}{g(t)}$ denotes the rescaled drift difference.

Given a fixed control path $\mathbf{u}_{t,\tau} := \{u(x_s, s)\}_{s=t}^{t+\tau}$, the expectation and variance of the log-weight are

$$\mathbb{E}\left[\log w_\tau^{\mathrm{GR}}(\mathbf{x}_{t,\tau}) \mid \mathbf{u}_{t,\tau}\right] = -\frac{1}{2} \int_t^{t+\tau} \|u(x_s, s)\|^2 \, \mathrm{d}s,$$

$$\mathrm{Var}\left(\log w_\tau^{\mathrm{GR}}(\mathbf{x}_{t,\tau}) \mid \mathbf{u}_{t,\tau}\right) = \int_t^{t+\tau} \|u(x_s, s)\|^2 \, \mathrm{d}s.$$

We now consider a time-discretized version of the variance over an interval $[t, t + \tau]$ with $N$ steps of size $\Delta t = \tau / N$. Denote the discretized control as $\mathbf{u}_{t,\tau}^{0:N-1} := \{u(x^k, t^k)\}_{k=0}^{N-1}$. Exponentiating this log-weight, the conditional variance of the log-weight becomes

$$\mathrm{Var}\left(w_\tau^{\mathrm{GR}}(\mathbf{x}_{t,\tau}^{0:N}) \mid \mathbf{u}_{t,\tau}^{0:N-1}\right) = \exp\left(\sum_{k=0}^{N-1} \|u(x^k, t^k)\|^2 \Delta t\right).$$

This reveals that the variance of the Girsanov weight grows exponentially with both trajectory duration $\tau$ and the dimension of control magnitude $u$, where $\|u(\cdot, t)\|^2$ scales with the dimension $d$.

## B  RATIO OF TRANSITION DENSITY

Let $\mathbf{x}_{t,\tau} = \{x_s\}_{s=t}^{t+\tau}$ denote the path space of continuous trajectories. We consider two probability measures, which are $\mu(\mathbf{x}_{t,\tau})$ (the original process defined in Eq. 1) and $\tilde{\mu}(\mathbf{x}_{t,\tau})$ (the perturbed process with drift term $\tilde{f}(\cdot, t)$). For any bounded measurable test function $O : \mathbb{R}^d \times \mathbb{R}^d \to \mathbb{R}$,

$$
\begin{aligned}
\mathbb{E}_{\rho_\tau(x_t, x_{t+\tau})}[O(x_t, x_{t+\tau})] &= \int O(x_t, x_{t+\tau})\rho_\tau(x_t, x_{t+\tau})\mathrm{d}x_t \mathrm{d}x_{t+\tau} \\
&= \int O(x_t, x_{t+\tau})\delta(X_t - x_t)\delta(X_{t+\tau} - x_{t+\tau})\mathrm{d}\mu(\mathbf{x}_{t,\tau}) \\
&= \int O(x_t, x_{t+\tau})\delta(X_t - x_t)\delta(X_{t+\tau} - x_{t+\tau})\frac{\mathrm{d}\mu}{\mathrm{d}\tilde{\mu}}(\mathbf{x}_{t,\tau})\mathrm{d}\tilde{\mu}(\mathbf{x}_{t,\tau}) \\
&= \int O(x_t, x_{t+\tau})\tilde{\rho}_\tau(x_t, x_{t+\tau})\frac{\mathrm{d}\mu}{\mathrm{d}\tilde{\mu}}(\mathbf{x}_{t,\tau})\mathrm{d}\tilde{\mu}(\mathbf{x}_{t,\tau}) \\
&:= \mathbb{E}_{\tilde{\rho}_\tau(x_t, x_{t+\tau})}[w_\tau(x_t, x_{t+\tau})O(x_t, x_{t+\tau})],
\end{aligned}
$$

where $w_\tau(x_t, x_{t+\tau}) = \mathbb{E}_{\tilde{\mu}}\left[\frac{\mathrm{d}\mu}{\mathrm{d}\tilde{\mu}}(\mathbf{x}_{t,\tau}) \mid X_t = x_t, X_{t+\tau} = x_{t+\tau}\right]$. Here, $\rho_\tau(x_t, x_{t+\tau})$, $\tilde{\rho}_\tau(x_t, x_{t+\tau})$ denote the joint marginal distributions under the original and perturbed processes respectively.

This formulation suggests that the ratio of transition densities $w_\tau(x_t, x_{t+\tau})$ can be estimated by Girsanov reweighting. However, beyond the well-known issue of rapidly growing variance in Appendix A, Girsanov reweighting computes weights tied to specific trajectories $\mathbf{x}_{t,\tau}$, whereas the desired transition ratio $w_\tau(x_t, x_{t+\tau})$ corresponds to an expectation over paths connecting the given endpoints.

## C  CONSISTENCY OF THE APPROXIMATE $\rho_{k\tau}(x, y)$

For the given iteration $k > 1$, let $Z := X_{t+(k-1)\tau}$ and denote the short lag-$\tau$ path $\mathbf{x}_{t+(k-1)\tau, \tau} = \{x_s\}_{s=t+(k-1)\tau}^{t+k\tau}$. For any bounded measurable $O : \mathbb{R}^d \times \mathbb{R}^d \to \mathbb{R}$, an unbiased estimation of

transition properties can be obtained by

$$\mathbb{E}_{\rho_{k\tau}(x_t, x_{t+k\tau})} [O(x_t, x_{t+k\tau}))] = \mathbb{E}_{\tilde{\rho}_{(k-1)\tau}(x_t, z), \, \tilde{\mu}(\mathbf{x}_{t+(k-1)\tau, \tau}|z)} [c_t O(x_t, x_{t+k\tau})],$$

where $c_t = w_{(k-1)\tau}(x_t, z) w_\tau^{\mathrm{GR}}(\mathbf{x}_{t+(k-1)\tau, \tau})$. Here, $w_{(k-1)\tau}(x_t, z)$ is the marginal weights inherited from the previous iteration, and $w_\tau^{\mathrm{GR}}(\mathbf{x}_{t+(k-1)\tau, \tau})$ is the pathwise Girsanov reweighting introduced in Section 3.2.

In practice we approximate this expectation by Monte Carlo. Extract all paired data $\{(x^i, \mathbf{x}_{i+(k-1)\tau, \tau}^{0:N})\}_{i=1}^M$ under $\tilde{\mu}$ from either a single long trajectory or multiple trajectories, where each short segment $\mathbf{x}_{i+(k-1)\tau, \tau}^{0:N}$ connects the intermediate endpoints $(z^i, y^i)$ with lag $\tau$, (thus the total lag between $x^i$ and $y^i$ is $k\tau$). With an assigned pathwise weight $c^i = w_{(k-1)\tau}(x^i, z^i) w_\tau^{\mathrm{GR}}(\mathbf{x}_{i+(k-1)\tau, \tau}^{0:N})$, the expectation can be estimated by

$$\mathbb{E}_{\rho_{k\tau}(x_t, x_{t+k\tau})} [O(x_t, x_{t+k\tau})] \approx \sum_{i=1}^M \frac{c^i O(x^i, y^i)}{\sum_{i=1}^M c^i}.$$

The estimation error approaches zero when $w_{(k-1)\tau}$ is accurate and $M \to \infty$.

*Proof:*

By Chapman–Kolmogorov (Hachigian, 1963), we have

$$\mathbb{E}_{\rho_{k\tau}(x_t, x_{t+k\tau})} [O(x_t, x_{t+k\tau})] = \int \rho_{k\tau}(x_t, x_{t+k\tau}) O(x_t, x_{t+k\tau}) \mathrm{d}x_t \mathrm{d}x_{t+k\tau}$$

$$= \int \rho_{(k-1)\tau}(x_t, z) p(x_{t+k\tau} \mid z) O(x_t, x_{t+k\tau}) \mathrm{d}z \mathrm{d}x_t \mathrm{d}x_{t+k\tau}$$

$$= \int \frac{\rho_{(k-1)\tau}(x_t, z) p(x_{t+k\tau} \mid z)}{\tilde{\rho}_{(k-1)\tau}(x_t, z) \tilde{p}(x_{t+k\tau} \mid z)} \cdot$$
$$\tilde{\rho}_{(k-1)\tau}(x_t, z) \tilde{p}(x_{t+k\tau} \mid z) O(x_t, x_{t+k\tau}) \mathrm{d}z \mathrm{d}x_t \mathrm{d}x_{t+k\tau}.$$

According to Section 4.1, $\frac{p(x_{t+k\tau}|z)}{\tilde{p}(x_{t+k\tau}|z)} = \mathbb{E}_{\tilde{\mu}(\mathbf{x}_{t+(k-1)\tau, \tau})} \left[ w_\tau^{\mathrm{GR}}(\mathbf{x}_{t+(k-1)\tau, \tau}) \mid Z = z, X_{t+k\tau} = x_{t+k\tau} \right]$ and $w_{(k-1)\tau}(x_t, z) = \frac{\rho_{(k-1)\tau}(x_t, z)}{\tilde{\rho}_{(k-1)\tau}(x_t, z)}$. It yields

$$\mathbb{E}_{\rho_{k\tau}(x_t, x_{t+k\tau})} [O(x_t, x_{t+k\tau})] = \int \tilde{\rho}_{(k-1)\tau}(x_t, z) \tilde{p}(x_{t+k\tau} \mid z) w_{(k-1)\tau}(x_t, z) w_\tau^{\mathrm{GR}}(\mathbf{x}_{t+(k-1)\tau, \tau})$$

$$O(x_t, x_{t+k\tau}) \tilde{\mu}(\mathrm{d}\mathbf{x}_{t+(k-1)\tau, \tau} \mid z, x_{t+k\tau}) \mathrm{d}z \mathrm{d}x_t \mathrm{d}x_{t+k\tau}$$

$$= \int \tilde{\rho}_{(k-1)\tau}(x_t, z) \tilde{\mu}(\mathrm{d}\mathbf{x}_{t+(k-1)\tau, \tau} \mid z)$$

$$w_{(k-1)\tau}(x_t, z) w_\tau^{\mathrm{GR}}(\mathbf{x}_{t+(k-1)\tau, \tau}) O(x_t, x_{t+k\tau}) \mathrm{d}z \mathrm{d}x_t$$

$$:= \mathbb{E}_{\tilde{\rho}_{(k-1)\tau}(x_t, z), \, \tilde{\mu}(\mathbf{x}_{t+(k-1)\tau, \tau}|z)} [c_t O(x_t, x_{t+k\tau})].$$

Due to the fact $\mathbb{E}_{\rho_{k\tau}(x_t, x_{t+k\tau})} [O(x_t, x_{t+k\tau})] = \mathbb{E}_{\tilde{\rho}_{k\tau}(x_t, x_{t+k\tau})} [w_{k\tau}(x_t, x_{t+k\tau}) O(x_t, x_{t+k\tau})]$, we can also prove $w_{k\tau}(x_t, x_{t+k\tau}) = \mathbb{E}_{\tilde{\rho}_{(k-1)\tau}(x_t, z), \, \tilde{\mu}(\mathbf{x}_{t+(k-1)\tau, \tau}|z)} [c_t]$, which provides the theoretical consistency across all lag scales.

Let $\{(x^i, y^i)\}_{i=1}^M$ be endpoint pairs sampled under $\tilde{\mu}$ and their pathwise weights drawn from $c^i = w_{(k-1)\tau}(x^i, z^i) w_\tau^{\mathrm{GR}}(\mathbf{x}_{i+(k-1)\tau, \tau}^{0:N})$. By the strong law of large numbers for Markov chains with ergodic assumption (Breiman, 1960),

$$\sum_{i=1}^M \frac{c^i O(x^i, y^i)}{\sum_{i=1}^M c^i} \xrightarrow{\text{a.s.}} \mathbb{E}_{\tilde{\rho}_{(k-1)\tau}(x_t, z), \, \tilde{\mu}(\mathbf{x}_{t+(k-1)\tau, \tau}|z)} [c_t O(x_t, x_{t+k\tau})] = \mathbb{E}_{\rho_{k\tau}(x_t, x_{t+k\tau})} [O(x_t, x_{t+k\tau}))].$$

# D TRAINING ALGORITHM

To summarize the MGR procedure, we present the full training and evaluation process in Algorithm 1. At each iteration indexed by lag time $k\tau$, the goal is to train a classifier to estimate the

marginal density ratio $w_{k\tau}(x_t, x_{t+k\tau})$ between the original and perturbed transition distributions. This is achieved by constructing pathwise weights $c_t$ that combine the short-time Girsanov weight $w_\tau^{\text{GR}}$ with the model prediction from the previous iteration $w_{(k-1)\tau}$. The resulting $w_{k\tau}$ is then used in the next iteration, allowing the model to progressively extend from short to long lag times.

---

**Algorithm 1** Marginal Girsanov Reweighting (MGR)

---

**Require:** Simulation trajectory $\{x_t\}_{t=0}^T$ from perturbed dynamics, lagtime $\tau$, the maximum training iteration $K$, learining rate $\eta$.
 1: **for** $k = 1$ to $K$ **do**
    *Step 1:* **Collect paired data**
 2:    Collect training pairs $\{(x_t, x_{t+k\tau})\}_{t=0}^{T-k\tau}$;
    *Step 2:* **Compute pathwise weights**
 3:    Compute Girsanov weight $w_\tau^{\text{GR}}(\mathbf{x}_{t+(k-1)\tau,\tau}^{0:N})$ using Eq. 2;
 4:    Compute pathwise weight:
 5:    **if** $k = 1$ **then**
 6:        $c_t = w_\tau^{\text{GR}}(\mathbf{x}_{t+(k-1)\tau,\tau}^{0:N})$;
 7:    **else**
 8:        $c_t = w_{(k-1)\tau}(x_t, x_{t+(k-1)\tau})w_\tau^{\text{GR}}(\mathbf{x}_{t+(k-1)\tau,\tau}^{0:N})$;
 9:    **end if**
    *Step 3:* **Train density ratio estimator**
10:    **for** each training epoch **do**
11:        **for** each minibatch $\{(x^{(i)}, y^{(i)}, c^{(i)})\}_{i=1}^B$ drawn from $\{(x_t, x_{t+k\tau}, c_t)\}_{t=0}^{T-k\tau}$ **do**
12:            Compute weighted binary classification loss with normalized weight $c^{(i)}$:

$$\mathcal{L}(\theta) = -\frac{1}{B}\sum_{i=1}^B \left[ c^{(i)} \log h_\theta(x^{(i)}, y^{(i)}) + \log(1 - h_\theta(x^{(i)}, y^{(i)})) \right];$$

13:            Update parameters: $\theta \leftarrow \theta - \eta \cdot \nabla_\theta \mathcal{L}(\theta)$;
14:        **end for**
15:    **end for**
16:    Update the marginal ratio $w_{k\tau}(x_t, x_{t+k\tau}) = \frac{h_\theta^*(x_t, x_{t+k\tau})}{1-h_\theta^*(x_t, x_{t+k\tau})}$;
17: **end for**
18: **return** Marginal ratio $w_{k\tau}(x_t, x_{t+k\tau})$ by model

---

# E EXPERIMENTAL DETAILS

## E.1 MOLECULAR DYNAMICS

In this section, we provide further details on the experimental setup, evaluation metrics, and implementation of MGR in the molecular dynamics (MD) setting described in Section 5.1.

For each observed transition pair $(x_t, x_{t+\tau})$, a marginal weight $w_\tau(x_t, x_{t+\tau})$ is assigned. The cross-correlation then becomes

$$
\begin{aligned}
C_{ij}(\tau) &= \mathbb{E}_\mu \left[ \mathbf{1}_{B_i}(x_t)\mathbf{1}_{B_j}(x_{t+\tau}) \right] \\
&= \mathbb{E}_{\tilde{\mu}} \left[ w_\tau(x_t, x_{t+\tau})\mathbf{1}_{B_i}(x_t)\mathbf{1}_{B_j}(x_{t+\tau}) \right] \\
&\approx \frac{\sum_t w_\tau(x_t, x_{t+\tau})\mathbf{1}_{B_i}(x_t)\mathbf{1}_{B_j}(x_{t+\tau})}{\sum_t w_\tau(x_t, x_{t+\tau})}.
\end{aligned}
$$

Here $w_\tau(x_t, x_{t+\tau})$ is estimated by MGR in our model. In the GR baseline, $w_\tau(x_t, x_{t+\tau})$ is computed via pathwise Girsanov reweighting. Once the corrected matrix is constructed, standard MSM analysis can be performed.

We adopt the following metrics, as described in the main text, to assess the performance of MGR:

**Effective Sample Size (ESS):** ESS is a popular metric to measure the variance of weights and is regarded as a metric for quantifying goodness of reweighting ratio on importance sampling. Denoting $\{w_{k\tau}(x^i, y^i)\}_{i=1}^M$ the weights assigned to data pairs, we compare the relative ESS, defined

as

$$\text{rESS}_{k\tau} = \frac{1}{M}\text{ESS}_{k\tau} = \frac{\left(\sum_t w_{k\tau}(x_t, x_{t+\tau})\right)^2}{M\sum_t w_{k\tau}(x_t, x_{t+\tau})^2}.$$

The relative ESS value lies in the interval $(0, 1]$, and a higher value indicates reduced variance and greater weight stability.

**Implied Timescales (ITS):** ITS is obtained by eigendecomposition of the reweighted transition probability matrix. We evaluate both individual timescales and their cumulative behavior as a function of lag time $\tau$, comparing with the unbiased reference to assess how well dynamical relaxation is captured.

For a lagtime $\tau$ transition matrix, let $1 = \lambda_1(\tau) > \lambda_2(\tau) \geq \lambda_3(\tau) \geq \cdots$ be the leading eigenvalues. Each eigenvalue defines an implied timescale

$$t_i(\tau) = -\frac{\tau}{\ln \lambda_i(\tau)}.$$

Each $t_i$ measures the relaxation time of a distinct slow dynamical mode: larger $t_i$ means slower decay (stronger metastability).

Beyond inspecting individual $t_i$, it is important to track an aggregate relaxation measure over the first $m$ slow modes:

$$S_m(\tau) := \sum_{i=2}^{m+1} \lambda_i(\tau) = \sum_{i=2}^{m+1} \exp\left(-\frac{\tau}{t_i(\tau)}\right).$$

This quantity decays exponentially with lagtime $\tau$, because $t_i(\tau)$ of a slow mode is usually approximately constant across $\tau$. Accurate tracking of this decay indicates that the model captures the intrinsic relaxation behavior of the dynamics (i.e., the spectrum of slow processes). In practice, we report: (i) several dominant ITS $t_i(\tau)$ to assess mode-wise accuracy, and (ii) the aggregate $S_m(\tau)$ over the first $m$ ITS, to summarize overall slow relaxation captured by the estimated transition matrix.

**Dominant eigenfunctions and stationary distribution:** For a lagtime $\tau$ transition matrix $P$, the stationary distribution $\pi$ satisfies $\pi^\top P = \pi^\top$ and represents the long-time equilibrium. Equivalently, $\pi$ is the first left eigenvector $\phi_1$ of $P$ with eigenvalue 1. The remaining leading eigenfunctions (right eigenfunctions $\{\psi_2, \psi_3, ...\}$, left eigenfunctions $\{\phi_2, \phi_3, ...\}$) with eigenvalues $1 > \lambda_2 \geq \lambda_3 \geq \cdots$ encode the slow dynamical modes: their sign pattern and level sets partition state space into metastable regions and provide approximate coordinates. Accurate recovery of $\pi$ validates thermodynamics, and accurate recovery of $\{\psi_i, \phi_i\}$ validates kinetics.

### E.1.1 1 DIMENSIONAL FOUR WELL

We consider an overdamped Langevin dynamics on a one–dimensional four–well landscape,

$$\mathrm{d}X_t = -\nabla V(X_t)\,\mathrm{d}t + \sigma\,\mathrm{d}W_t,$$

where $V(x) = 4\left(x^8 + 0.8e^{-80x^2} + 0.2e^{-80(x-0.5)^2} + 0.5e^{-40(x+0.5)^2}\right)$, $\sigma = 1$.

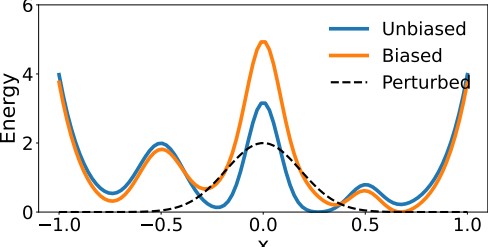 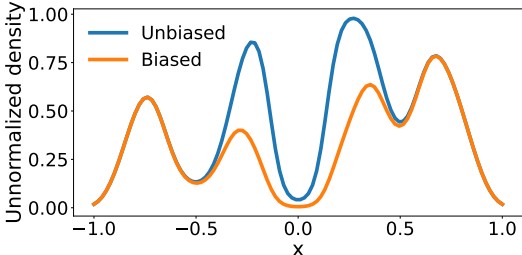

Figure S.1: Energy and density of 1D four-well system. **Left:** Energy profiles of unbiased potential $V(x)$, biased potential $\widetilde{V}(x) = V(x) + U(x)$, and the bias term $U(x)$. **Right:** Theoretical unnormalized probability of the system.

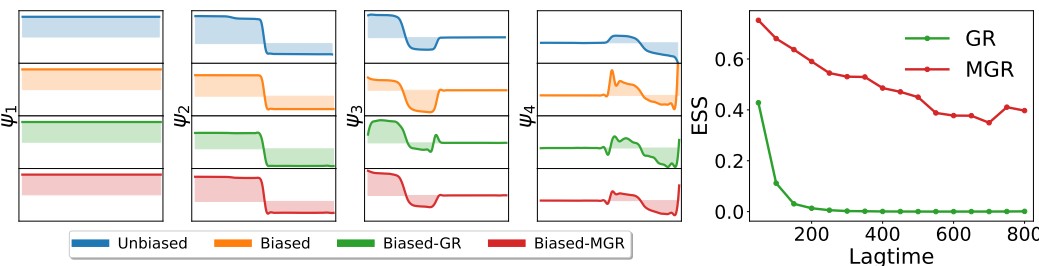

Figure S.2: **Left:** Dominant right ($\{\psi_i\}$) eigenfunctions of the transition matrix at lagtime $300\Delta t$. **Right:** Relative ESS as a function of lag time comparing GR and MGR.

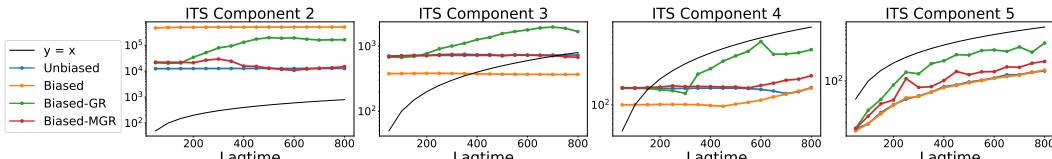

Figure S.3: The first four implied timescales (ITS) versus lag time. Each panel reports $t_i(\tau)$ for mode $i = 2, ..., 5$ (log scale). MGR tracks the unbiased ITS with less fluctuation, while GR deviates at long lagtimes.

To accelerate barrier crossings between the two intermediate wells, we introduce a perturbed potential

$$\mathrm{d}X_t = -\nabla \widetilde{V}\left(X_t\right)\mathrm{d}t + \sigma\,\mathrm{d}W_t,$$

where $\widetilde{V}(x) = V(x) + U(x)$, $U(x) = 2e^{-kx^2}$, $k = 15$. Illustrations of the potentials and stationary densities are provided in Figure S.1.

We use the Euler–Maruyama scheme with time step $\Delta t = 10^{-3}$. Each run is integrated up to $T = 10,000$, yielding $T/\Delta t = 10^7$ samples per trajectory. We initialize at $X_0 = 0$ and simulate a single trajectory. During the perturbed run, we record the discrete time Girsanov log-weights according to Eq. 2, i.e.,

$$\log w_{\Delta t}^{\mathrm{GR}}(x) = -\nabla U(x)\sqrt{\Delta t}\,\xi - \frac{\Delta t}{2}\left(\nabla U(x)\right)^2,$$

where $\xi$ is the corresponding noise in the simulation.

We set $\tau = 50\Delta t$ as the reference short GR lagtime during training. At this short lagtime, the Girsanov weights are numerically stable, with ESS $\approx 0.43$. Guidelines for selecting a suitable short lag $\tau$ are provided in Appendix F.2. In each training iteration $k$, we normalize the pathwise weights

$$c_t = \frac{w_{(k-1)\tau}(x_t, x_{t+(k-1)\tau})\,w_\tau^{\mathrm{GR}}(\mathbf{x}_{t+(k-1)\tau,\tau}^{0:N})}{\frac{1}{M}\sum_t\left(w_{(k-1)\tau}(x_t, x_{t+(k-1)\tau})\,w_\tau^{\mathrm{GR}}(\mathbf{x}_{t+(k-1)\tau,\tau}^{0:N})\right)}, \tag{S.1}$$

where $M$ is the number of total paired data in our training dataset. After training, we obtain the estimated marginal ratio $w_{k\tau}(x_t, x_{t+k\tau}) = \frac{h_\theta^*(x_t, x_{t+k\tau})}{1-h_\theta^*(x_t, x_{t+k\tau})}$ for reweighting.

The reweighted transition counts yield an approximation of the unbiased transition matrix, enabling standard MSM analysis. The dominant right eigenfunctions and the relative Effective sample size (ESS) as a function of lag time are shown in Figure S.2 and Figure S.3 reports the first 4 implied timescales individually. Furthermore, Figure S.4 displays the density plot of the transfer operator, which can also be reweighted by MGR ratio

$$\rho_{k\tau}(x, y) = \rho(x)p_{k\tau}(x, y)$$
$$= e^{-V(x)}\tilde{\rho}(x)w_{k\tau}(x, y)\tilde{p}_{k\tau}(x, y)$$
$$= e^{-V(x)}w_{k\tau}(x, y)\tilde{\rho}_{k\tau}(x, y).$$

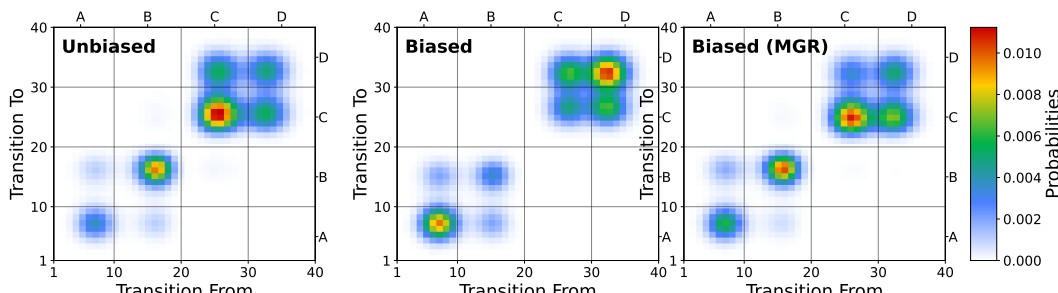

Figure S.4: Density plot of the transfer operator for Four Well, defined on the discrete spatial range $[0, 40]$ with a lag time of $300\Delta t$. Red indicates high transition probability, white zero transition probability.

In a four-well landscape, the first three slow kinetic modes are important. The slowest ITS (ITS component 2) corresponds to the global left–right rearrangement across the highest barriers. ITS component 3 and the third eigenfunctions in Figures 3, S.2 capture the exchanges between the two left wells (left-pair exchange), while ITS component 4 and associated eigenfunctions resolve the remaining transfer with the two right wells (right-pair exchange). The rest of ITS is small and unstable across lagtimes, reflecting fast relaxation rather than a meaningful slow process.

### E.1.2   ALANINE DIPEPTIDE

We performed all-atom MD simulations of acetyl-alanine-methylamide (Ac-A-NHMe, alanine dipeptide) in implicit water. The simulation was carried out with the OPENMM 8.2 simulation package (Eastman et al., 2023) at 300K. The system employed the Amber14 force field with OBC2 implicit water ("`amber14-all.xml`", "`implicit/obc2.xml`"). Dynamics were propagated with an Underdamped Langevin integrator with time step 2fs. The aggregated simulation time was $1\mu$s. Coordinates and Girsanov reweighting (GR) factors were saved every 20 steps (40fs) using the Girsanov-enabled OPENMM implementation (Schafer & Keller, 2024).

For enhanced sampling, we applied a dihedral bias to the backbone torsions $\phi$ and $\psi$. The unbiased dihedral potentials were

$$
\begin{aligned}
V(\phi) &= 0.27\cos(2\phi) + 0.42\cos(3\phi), \\
V(\psi) &= 0.45\cos(\psi - \pi) + 1.58\cos(2\psi - \pi) + 0.44\cos(3\psi - \pi),
\end{aligned}
$$

and the perturbation was a quadratic restraint

$$
U(\phi, \psi) = \frac{1}{2}\kappa_\phi\phi^2 + \frac{1}{2}\kappa_\psi\psi^2, \quad \kappa_\phi = \kappa_\psi = 1,
$$

so that the biased potential is $\widetilde{V}(\cdot) = V(\cdot) + U(\phi, \psi)$. The perturbation lowers the energy barrier in targeted regions of the $(\phi, \psi)$ free-energy surface, thereby facilitating transitions among metastable basins (Figure S.5), but it also distorts equilibrium and kinetics.

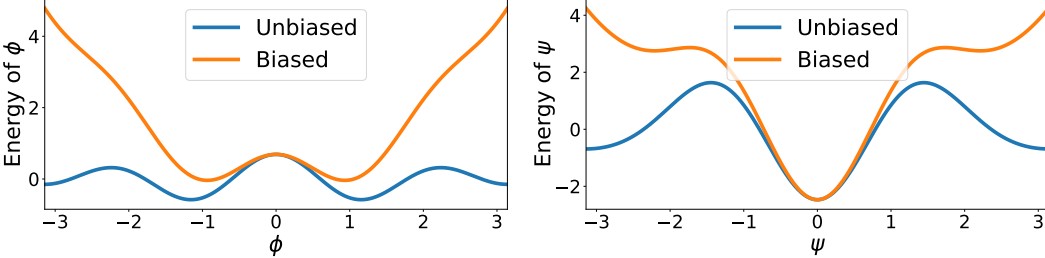

Figure S.5: Alanine dipeptide dihedral energy function. **Left:** $\phi$ torsion: unbiased energy $V(\phi)$ and biased function $V(\phi) + \frac{1}{2}\kappa_\phi\phi^2$. **Right:** $\psi$ torsion: unbiased energy $V(\psi)$ and biased function $V(\psi) + \frac{1}{2}\kappa_\psi\psi^2$.

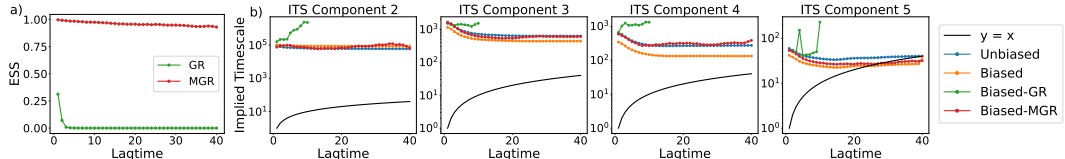

Figure S.6: Relative ESS and the first four implied timescales (ITS) versus lag time in Alanine dipeptide. a) Relative ESS as a function of lag time comparing GR and MGR. b) Each panel reports $t_i(\tau)$ for mode $i = 2, ..., 5$ (log scale). MGR tracks the unbiased ITS with less fluctuation, while GR shows huge deviation. GR totally fails to construct a valid MSM beyond lag time longer than $10\tau = 400$fs.

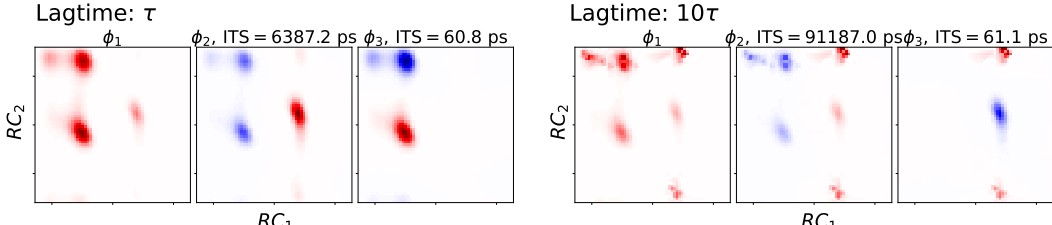

Figure S.7: Dominant eigenfunctions at lagtime $\tau$ and $10\tau$ of biased trajectories, reweighting by GR.

We set $\tau = 40$fs as the reference short GR lagtime during training. At this short lagtime, the Girsanov weights are numerically stable, with ESS $\approx 0.31$. In each training iteration $k$, we normalize the pathwise weights according to Eq. S.1.

The reweighted transition counts yield an approximation of the unbiased transition matrix, enabling standard MSM analysis. For alanine dipeptide, we construct the MSM in the key reaction coordinates $(\phi, \psi)$ space. In our model, we likewise use only two angles as inputs to the ratio estimator, which yields satisfactory performance.

The results are shown in Figure 4, and Figure S.6 further reports ESS and the first 4 ITS individually. As the lagtime increases, the variance of pathwise GR grows rapidly and the effective sample size collapses. Beyond $10\tau$, GR fails to produce a stable transition matrix and thus cannot construct a valid MSM. For illustration, we plot the dominant eigenfunctions at $\tau$ and $10\tau$ reweighted by GR in Figure S.7. At $\tau$, GR yields corrected modes, but at $10\tau$ the modes become distorted and metastable partitions blur. In contrast, MGR retains stable eigenfunctions across lags as shown in Figure 4 that closely match the unbiased reference, consistent with its higher ESS and accurate ITS.

### E.2 BAYESIAN INFERENCE OF SDE PARAMETERS

Consider a $d$-dimensional diffusion process that satisfies the following SDE:

$$\mathrm{d}X_t = f(X_t, t; \theta)\mathrm{d}t + g(t)\mathrm{d}W_t, \quad X_0 \sim p_0,$$

where $X_t \in \mathbb{R}^d$ denotes the state at time $t$, $f(\cdot, t; \theta)$ is the drift depending on the unknown parameter vector $\theta$, $g(\cdot)$ is the diffusion coefficient, and $W_t$ is a standard Wiener process. Suppose we observe noisy measurements $\mathbf{y} = \{y(t = 0) = y^0, y^1, ..., y(t = T) = y^K\}$ at equally spaced time intervals with lag time $T/K$. The goal is to infer the unknown parameter vector $\theta$ from these observations. Within the Bayesian framework, we assign a prior distribution $p(\theta)$ and aim to recover the posterior

$$p(\theta \mid \mathbf{y}) \propto p(\mathbf{y} \mid \theta) p(\theta).$$

The main difficulty lies in computing the likelihood $p(\mathbf{y}|\theta)$. Since the process $\mathbf{X} = \{x(t = 0) = x^0, x^1, ..., x(t = T) = x^K\}$ is governed by the SDE with parameter $\theta$, the likelihood requires integrating over all possible trajectories,

$$p(\mathbf{y} \mid \theta) = \int p(\mathbf{y} \mid \mathbf{X}, \theta) p(\mathbf{X} \mid \theta) \mathrm{d}\mathbf{X},$$

which is generally intractable.

Instead of simulating the SDE for every candidate $\theta$, we set a reference parameter $\theta_0$. We then generate reference trajectories under $\theta_0$ and compute the ratio of likelihood ratio between candidates $\theta$ to $\theta_0$. According to Girsanov reweighting in Section 3.2, the ratio $\frac{\mathrm{d}p(\mathbf{X}|\theta)}{\mathrm{d}p(\mathbf{X}|\theta_0)}$ can be recorded during the simulation, where the drift difference in Eq. 2 is $f(X_t, t; \theta) - f(X_t, t; \theta_0)$.

Through the MGR model introduced in Section 4, we approximate the ratio with the given observation pair $(y_i, y_{i+1})$

$$w_{k\tau}^\theta(\mathbf{y}) = \frac{p(y^{0:K}|\theta)}{p(y^{0:K}|\theta_0)} = \Pi_i \frac{p(y^{i+1}|y^i; \theta)}{p(y^{i+1}|y^i; \theta_0)} = \Pi_i w_{k\tau}^\theta(y_i, y_{i+1}). \tag{S.2}$$

Consequently, posterior expectations can be evaluated by importance sampling using observations under $\theta_0$:

$$
\begin{aligned}
\mathbb{E}[\theta \mid \mathbf{y}] &= \int \theta\, p(\theta \mid \mathbf{y})\mathrm{d}\theta \\
&= \int \theta\, \frac{p(\mathbf{y} \mid \theta)p(\theta)}{p(\mathbf{y})}\mathrm{d}\theta \\
&= \int \theta\, \frac{p(\mathbf{y} \mid \theta)}{p(\mathbf{y} \mid \theta_0)} \frac{p(\mathbf{y} \mid \theta_0)p(\theta)}{p(\mathbf{y})}\mathrm{d}\theta \\
&= \frac{\int \theta \frac{p(\mathbf{y}|\theta)}{p(\mathbf{y}|\theta_0)} p(\theta)p(\mathbf{y} \mid \theta_0)\, d\theta}{\int \frac{p(\mathbf{y}|\theta)}{p(\mathbf{y}|\theta_0)} p(\theta)p(\mathbf{y} \mid \theta_0)\, d\theta} \\
&\approx \frac{\sum_\theta \theta\, w_{k\tau}^\theta(\mathbf{y})p(\theta)}{\sum_\theta w_{k\tau}^\theta(\mathbf{y})p(\theta)}.
\end{aligned}
\tag{S.3}
$$

For the baseline particle marginal Metropolis–Hastings (PMMH) algorithm (Golightly & Wilkinson, 2008; Hoffman et al., 2014) and a variational inference (VI) approach (Ghosh et al., 2022). We use the default setting in their original paper.

### E.2.1 GRAPH ORNSTEIN-UHLENBECK

We consider a two dimensional Graph Ornstein-Uhlenbeck (OU) process as introduced in Section 5.2.1. The Euler–Maruyama scheme with time step $\Delta t = 10^{-3}$ is used to generate a trajectory and the simulation is integrated up to $T = 30$, yielding $M = T/\Delta t = 3 \times 10^4$ samples. A set of 30 evenly spaced values constitute the observations $\mathbf{y}$, and each observation is corrupted with Gaussian noise of $\sigma = 0.5$, shown in Figure S.8. We consider 15 candidate values of $\theta^1$ in the range $[0.1, 1.5]$ with increments of 0.1, and 15 values of $\theta^2$ in the range $[0.5, 7.5]$ with increments of 0.5. The resulting grid consists of 225 parameter combinations, and for each of them we compute the corresponding Girsanov reweighting during simualtion.

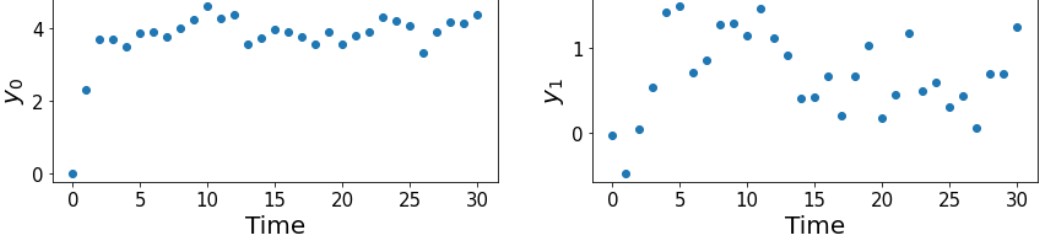

Figure S.8: Observations of Graph Ornstein-Uhlenbeck.

### E.2.2 STOCHASTIC LOTKA-VOLTERRA

We consider a two dimensional stochastic Lotka-Volterra system as introduced in Section 5.2.2. The Euler–Maruyama scheme with time step $\Delta t = 10^{-3}$ is used to generate a trajectory and the simulation is integrated up to $T = 50$, yielding $M = T/\Delta t = 5 \times 10^4$ samples. A set of 25 evenly spaced values from this path corrupted with Gaussian noise of $\sigma = 5$ constitute the observations $\mathbf{y}$,

shown in Figure S.9. We consider 15 candidate values of $\theta^1$ in the range $[0.3, 0.5]$ with increments of 0.014, and 15 values of $100 \times \theta^2$ in the range $[0.2, 0.4]$ with increments of 0.014. The resulting grid consists of 225 parameter combinations, and for each of them we compute the corresponding Girsanov reweighting during simualtion.

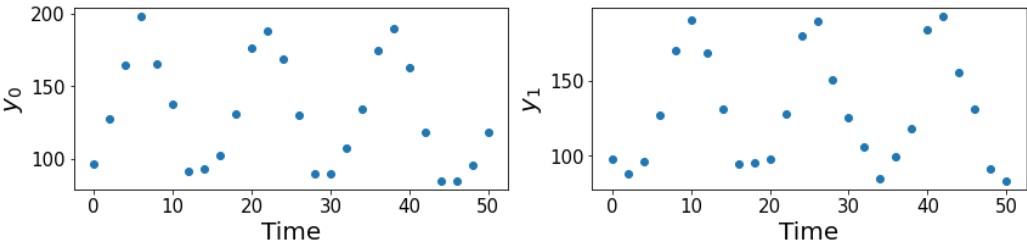

Figure S.9: Observations of Stochastic Lotka-Volterra.

### E.3 MODEL ARCHITECTURE

Our classifier estimator $h_\theta(\cdot, \cdot)$ in MGR is modeled by 3 layer MLPs, augmented with Fourier feature encodings (Tancik et al., 2020). We consider two types of encoding $\gamma(\mathbf{v})$, ($\mathbf{v} = [x_t, x_{t+k\tau}]$ is the concatenation of the paired data):

**Positional encoding:** $\gamma(\mathbf{v}) = [\sin(\mathbf{v}), \cos(\mathbf{v}), \sin(2\mathbf{v}), \cos(2\mathbf{v}), ..., \sin(B\mathbf{v}), \cos(B\mathbf{v})]$, where $B = 10$ denotes the scaling number.

**Gaussian encoding:** $\gamma(\mathbf{v}) = [\cos(B\mathbf{v}), \sin(B\mathbf{v})]$, where $B \in \mathbb{R}^{m \times 2d}$ is sampled from $\mathcal{N}(0, 10)$.

Both Fourier-feature variants converge faster during training and yield comparable estimation accuracy. Detailed comparison is reported in Appendix F.4. In our experiments, we adopt a positional-encoding MLP with ReLU activations.

## F ABLATION

### F.1 NEURAL RATIO ESTIMATION

Ratio estimation (Sugiyama et al., 2010) is a fundamental technique for comparing two distributions. Kernel moment matching, e.g. KMM (Gretton et al., 2009), matches all the moments with reproducing kernels, which is effective and computationally efficient. Probabilistic classification recasts ratio estimation as posteriors from a binary classifier (Menon & Ong, 2016), showing powerful fitting capability. Featurized classification with normalizing flows (Choi et al., 2021) further performs classification in a learned latent space, mitigating issues caused by large distributional discrepancies. Path-based methods (Choi et al., 2022; Yu et al., 2025) connect the two distributions via a continuous probability path and estimate the density ratio by integrating a learned time score. By constructing consecutive path distributions, it alleviates the problems caused by poor overlap between two densities.

However, unlike the standard setting with samples from both distributions, here we only have samples from one distribution plus reference weights linking two distributions. We therefore conducted minor adaptions to estimators below and compare their performance on Four well system. We also adapted the path-based method (Yu et al., 2025), but it exhibited numerical instability in our setting.

**Standard classifier (weighted BCE)** Following Section 4.3, we train a binary classifier on endpoint pairs $(x_t, x_{t+k\tau})$ with weighted cross-entropy in Eq. 6.

**Featurized classifier (weighted BCE)** We first map each paired sample to a latent representation $z_\phi = \Phi(x_t, x_{t+k\tau}; \phi)$, and then perform the classifier-based ratio estimation in this feature space. A joint training objective is adopted (Choi et al., 2021):

$$\mathcal{L}_{\text{joint}} = \alpha \mathcal{L}_{\text{BCE}}(\theta, \phi) + (1 - \alpha)\mathcal{L}_{\text{latent}}(\phi),$$

where $\mathcal{L}_{\text{BCE}}(\theta, \phi) = -\mathbb{E}_t\left[c_t \log h_\theta(z_\phi) + \log\left(1 - h_\theta(z_\phi)\right)\right]$ is the weighted binary cross entropy in latent space, $\mathcal{L}_{\text{latent}}(\phi)$ denotes the objective for optimizing encoder network, and $\alpha = 0.5$ is a hyperparameter.

Here, we consider two encoders to map data into latent space: (i) an invertible normalizing-flow encoder (Classifier-NF) (Choi et al., 2021), and (ii) a non-bijective MLP encoder (Classifier-MLP). Classifier–NF guarantees ratios computed in feature space are equivalent to those in input space, whereas Classifier-MLP uses more flexible, non-invertible networks at the cost of potential information loss. For Classifier-NF, we train the encoder by maximum likelihood loss $\mathcal{L}_{\text{latent}}^{\text{KL}}(\phi)$ (Dinh et al., 2016; Choi et al., 2021), and for Classifier-MLP, we minimize a sliced 2-wasserstein distance $\mathcal{L}_{\text{latent}}^{\text{Wass}}(\phi)$(Kolouri et al., 2018) between the latent variables and a standard Gaussian.

Comparative results are reported in Figure S.10, where featurized classifiers did not show measurable improvement over a standard classifier. We adopt the standard classifier in Section 4.3, which provides satisfying results. Alternative advanced ratio estimators, and their applications, require further investigation.

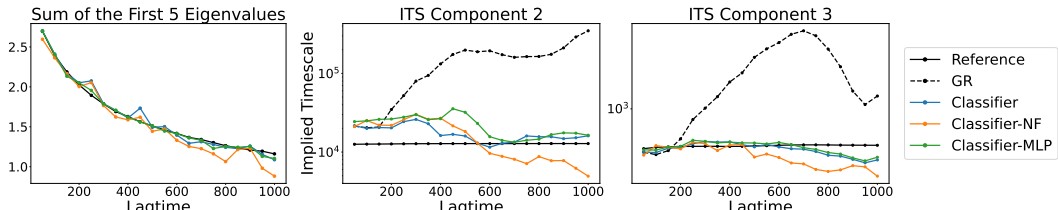

Figure S.10: Neural ratio estimators ablation on the 1D Four well system. **Left:** Sum of the first 5 eigenvalues as a function of lagtime across different ratio estimators. **Right:** The first 2 ITS as a function of lagtime across different ratio estimators.

### F.2 LAGTIME CHOSEN

Selecting the suitable short lag $\tau$ used in $w_\tau^{\text{GR}}$ is crucial for MGR. Large $\tau$ will inflate the variance of Girsanov weights and can destabilize training, while small $\tau$ will require many iterations to reach long timescales, accumulating approximation error and cost. As a practical rule, we recommend choosing $\tau$, which the relative ESS of the Girsanov weights falls in the range $0.3 \sim 0.5$. This strikes a balance between weight degeneracy and excessive iteration depth.

Take Four well system as an example. We train MGR with short-lag values $\tau \in \{25, 50, 75, 100, 150\}\Delta t$. The corresponding relative ESS of $w_\tau^{\text{GR}}$ is $\{0.73, 0.43, 0.24, 0.11, 0.06\}$. All other settings are kept identical. We then compare the dominant second implied timescale (ITS2) across evaluation lags in Figure S.11. It shows that the model trained under $\tau = 50\Delta t$ produces the most stable ITS2 and matched the unbiased (reference) result most closely. As $\tau$ increases, the discrepancy between the model results and the unbiased reference grows.

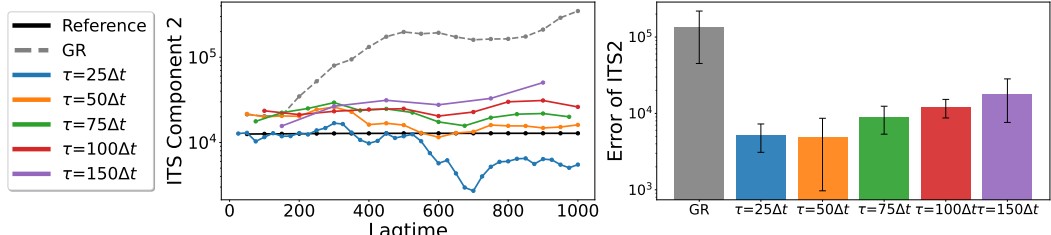

Figure S.11: Lagtime ablation on the 1D Four well system. **Left:** ITS2 as a function of lagtime for models trained with different short-lag values $\tau$. **Right:** Mean and standard deviation of the ITS2 error, aggregated over all evaluation lags for models trained with different short-lag values $\tau$.

### F.3 THE RANGE OF CANDIDATE PARAMETERS

A practical question in Bayesian inference is how to determine an appropriate range of candidate parameters around the reference value. In principle, the Girsanov reweighting estimator is valid for any parameter shift, but its variance grows rapidly when the reference and target dynamics exhibit

little overlap. Usually, the candidate parameters should typically be chosen as small perturbations around the reference value so that the path distributions remain close and the reweighting remains stable.

To investigate this effect, we performed an ablation study on the GraphOU and Lotka–Volterra (LV) SDEs using a broader candidate range. For each parameter value, we computed the effective sample size (ESS) of the raw Girsanov weights, which quantifies the degree of overlap between the reference and target dynamics. As shown in Figure S.12, the ESS values decay sharply as the parameter deviates from the reference, indicating that reliable reweighting is possible within a suitable perturbation region.

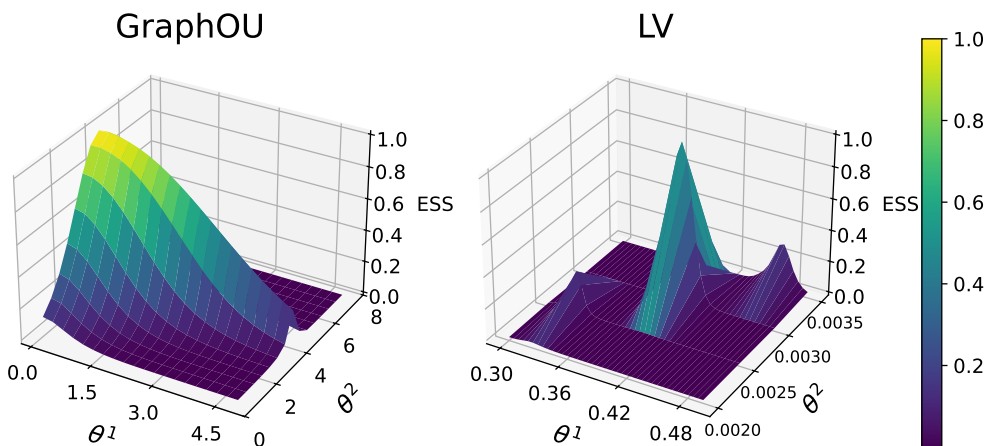

Figure S.12: ESS as a function of candidate parameters. ESS values computed from the raw Girsanov weights for the GraphOU (left) and LV (right) SDE examples.

### F.4 MODEL ARCHITECTURE

To demonstrate network efficiency, we compare a plain MLP, a positional encoding MLP, and a Gaussian encoding MLP on Four well system in Figure S.13. Both Fourier-feature models (Positional and Gaussian) converge faster than the plain MLP (Base). Three models show comparable estimation accuracy during evaluation.

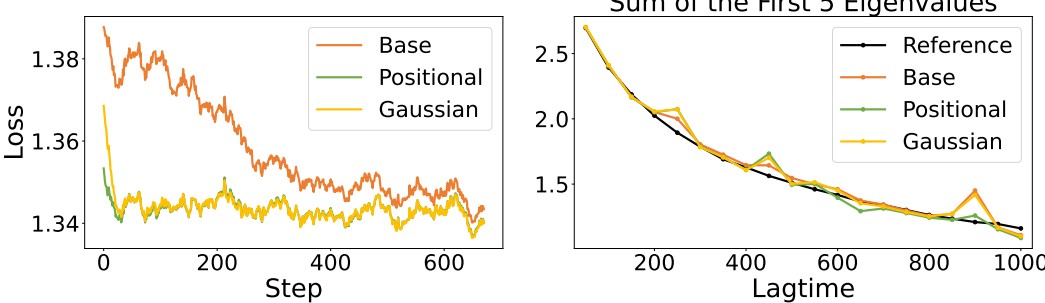

Figure S.13: Network ablation on the 1D Four well system. **Left:** Training curve as a function of training steps across different networks. **Right:** Sum of the first 5 eigenvalues as a function of lagtime across different networks.

