# OpenReview forum: "Marginal Girsanov Reweighting: Stable Variance Reduction via Neural Ratio Estimation"
_ICLR.cc/2026/Conference — Submitted to ICLR 2026_

### Official Review · Reviewer_Ji2i · 2025-10-24

**Soundness:** 2
**Presentation:** 2
**Contribution:** 3
**Rating:** 4
**Confidence:** 4

**Summary:**

In the study of molecular dynamics Monte Carlo simulations are frequently used to estimate the transition probabilities between metastable states. If the separating potential barrier is high such a simulation may take a long time. Modifying the potential V to V' can facilitate the climb over the energy barrier, but this perturbs the original statistics. Therefore a reweighting scheme is needed to compute the original transition probability from the one observed in the perturbed simulation. This reweighting factor can be written as an expectation, conditioned on the endpoints, of the Radon-Nikodym derivative of the original process driven by V with respect to the perturbed, simulated path driven by V'.
    For two processes sharing the diffusion coefficient Girsanov's theorem furnishes an expression for this Radon Nikodym derivative, and a method using a time-discretized version of the theorem in the study of molecular dynamics has been proposed, analyzed and used by Donati et al (2017). The paper to review now points out that the logarithm of the Girsanov derivative suffers from an increase of variance both with underlying dimension and length of trajectory. This problem is then addressed by applying Girsanov's theorem as in Donati et al (2017), but only over short time intervals and extending it to longer trajectories with a machine learning technique. There a classifier, trained to detect the different between (already trained) pairs of joint distributions from the original and the perturbed, is used recursively to extend the estimate of the likelyhood ratio, beginning from short time intervals where the Girsanov method is more reliable.

**Strengths:**

The proposed method seems quite original and the paper presents an impressive number of experiments to support the proposed method. The problem considered seems to be of considerable practical importance.

**Weaknesses:**

1) The authors have chosen to reweigh the transition probability using Girsanov theorem. They improve previous results based on this method using a clever machine learning trick. They show that they can obtain results that were impossible to get with previous methods. However, there is another way that was recently pointed to recover the same kind of information as done in the present work: instead of reweighting the individual transition probability, one can learn the eigenfunctions of the infinitesimal generator ("Devergne et al. Neurips 2024, https://openreview.net/forum?id=TGmwp9jJXl&noteId=txQ3k2U5vR) by reweighting accordingly using the umbrella sampling formula <O> = <O e^{\beta V}> /<e^{\beta V}>. It seems important to cite and compare to the method in this paper.

2) The description of the method is rather opaque. The back and forth between continuous and discretized processes as in appendix A (variance explodes) and other places is confusing.

3) The theoretical support for the method is very weak. In particular the authors fail to explain how the noise inherited from the discretized Girsanov method does not accumulate in the recursive training procedure.

4) The use of the strong law of large numbers in appendix C is not justified by any independence assumptions. For long trajectories, are there any mixing assumptions on the original and/or the perturbed process? Are the multiple trajectories at least independent?

**Questions:**

See weaknesses above.

In line 128 it should be $t \in [T - \tau]$ instead

At the end of line 207 shouldn't it be $(x_t,x_{t+k\tau})$?

(4) is not really an identity, because (3) is not.

Line 731: "We consider two probability on $\mu$..." delete \mu and parentheses.

---

> ### Author Response · Authors · 2025-11-20
>
> We thank the reviewer for their comments, the input has helped us better situate the our contribution in the light of recent contributions and helped us clarify the mathematical notation.
>
> &nbsp;
>
> ### W1.
> > The authors have chosen to reweigh the transition probability using Girsanov theorem. They improve previous results based on this method using a clever machine learning trick. They show that they can obtain results that were impossible to get with previous methods. However, there is another way that was recently pointed to recover the same kind of information as done in the present work: instead of reweighting the individual transition probability, one can learn the eigenfunctions of the infinitesimal generator ("Devergne et al. Neurips 2024, https://openreview.net/forum?id=TGmwp9jJXl\&noteId=txQ3k2U5vR) by reweighting accordingly using the umbrella sampling formula $O e^{\beta V}$. It seems important to cite and compare to the method in this paper.
>
> ### Response:
>
> We thank the reviewer for pointing us to the recent work of Devergne et al. (NeurIPS 2024). This method provides a Koopman-operator–based route to estimate long-time dynamical quantities, which recovers the unbiased eigenfunctions of the infinitesimal generator under biased equilibrium simualtions. We have added a citation and a discussion in Section 2. We also show the comparison results in the https://figshare.com/s/0ce048e7e9cec6d01ac7 .
>
> To clarify the relationship, we note that the approach of Devergne et al. differs from MGR in several fundamental aspects:
>
> **Dynamical regime**
>
> The baseline method is applicable only to overdamped Langevin dynamics. In contrast, MGR applies to both overdamped and underdamped Langevin systems, and more generally to any SDE with known drift/diffusion for which a Girsanov density exists. This broader applicability is essential for many MD systems where underdamped dynamics is the standard simulation protocol.
>
> **Equilibrium requirement**
>
> The baseline must rely on equilibrium (or near-equilibrium) sampling so that the umbrella-sampling bias term $e^{\beta V}$ correctly recovers the generator eigenfunctions. However, MGR imposes no equilibrium assumption: it reweights path measures directly via (variance-reduced) Girsanov ratios and therefore remains valid for trajectories generated under non-equilibrium trajectories. It is an important practical advantage in MD, where equilibrated samples usually be hard or expensive to obtain.
>
> **The learned object**
>
> The baseline method learns eigenfunctions of the generator and reweights dynamical quantities expressible in terms of these eigenfunctions. In contrast, MGR provides an explicit estimate of the path-wise likelihood ratio between biased and unbiased dynamics. This makes MGR applicable to a broader range of tasks beyond eigenfunction learning. For example, it can reweigh the transition operator density (shown in Appendix E.1.1, pages 18–19, marked in "blue") or performing path-space Bayesian inference, as shown in our experiments.
>
> Thanks again for the valuable suggestions.
>
> &nbsp;
>
> ### W2.
> >The description of the method is rather opaque. The back and forth between continuous and discretized processes as in appendix A (variance explodes) and other places is confusing.
>
> ### Response:
>
> We thank the reviewer for pointing out this source of confusion. Girsanov reweighting is inherently defined in continuous time: the Radon–Nikodym derivative between two SDE path measures is a continuous-time functional, and its variance properties are naturally analyzed at the level of the continuous diffusion process. For this reason, Appendix A begins with the continuous-time formulation, which provides the theoretical foundation and allows us to derive the variance expression for the exact Girsanov likelihood ratio.
>
> However, all practical simulations, e.g. in MD simualations, operate in discrete time. The form of Girsanov weights used in applications is therefore the discretized version of the continuous-time expression. Therefore, in the main text we present this discrete formula directly, as it is the implementation used in all experiments. In Appendix A, after establishing the continuous-time results, we explicitly present the corresponding discrete-time variance expression, which is the form employed in numerical computations.
>
> Thus, the back-and-forth between continuous and discrete formulations reflects the standard separation between (i) the theoretical SDE-level definition of Girsanov weights, and (ii) the practically used discretized estimator.

---

> > ### Author Response · Authors · 2025-11-20
> >
> > ### W3.
> > > The theoretical support for the method is very weak. In particular the authors fail to explain how the noise inherited from the discretized Girsanov method does not accumulate in the recursive training procedure.
> >
> > ### Response:
> >
> > We thank the reviewer for this theoretical justification question. We are not entirely certain what the reviewer refers to by "noise" here. Nonetheless, we address the issue from the following two perspectives, and we hope these clarifications resolve the reviewer’s concern. We would be glad to provide further clarification if we misunderstand the question.
> >
> > **About noise inherited from GR**
> >
> > The theoretical foundation of our method, established in Step 2 Section 4.2 (proof provided in Appendix C), describes how we reduce the variance from noise inherited from GR.
> > According to the following recursive identity for any bounded observable $O(x_t, x_{t+k\tau})$:
> >
> > $\mathbb{E}_{\rho_{k \tau}}[O]=\mathbb{E}_{\tilde{\rho}_{(k-1) \tau} \tilde{\mu}_{\tau}}\left[c_t O\right], \quad c_t=w_{(k-1) \tau}\left(x_t, x_{t+(k-1) \tau}\right) w_\tau^{\mathrm{GR}}\left(\mathbf{x}_{t+(k-1) \tau, \tau}\right) ,$ (the same equation as  in Step 2 of Section 4.2)
> >
> > we have $w_{k \tau}(x_t, x_{t+k\tau})=\mathbb{E}\left[c_t\mid X_t=x_t, X_{t+k\tau}=x_{t+k\tau}\right]$. It implies that the model is learning the expectation of pathwise ratio, which is also the ratio integrated over intermediate states along the trajectories in GR. As a result, the variance arising from the intermediate-path noise in GR is reduced. This constitutes the core innovation of MGR.
> >
> > Moreover, if $w_{(k-1) \tau}$ is an unbiased estimator of the marginal ratio at lag $(k-1) \tau$, then $w_{k \tau}$ is also the unbiased ratio at lag $k \tau$. The recursion therefore preserves consistency across all lag scales. We have added an additional clarification in the revised Appendix C marked in ``blue'' to make this reasoning more explicit.
> >
> > **About error inherited from discretized GR**
> >
> > When discretizing the SDE with the Euler scheme, the simulation inevitably introduces numerical errors. This is unavoidable, and the best we can do is to choose a small step sizes to ensure accurate simulation and GR computation. Before training the model, we assess the accuracy of the GR computation using the effective sample sizes (ESS) metric and then choose a reasonable $\tau$ as the short-lag GR interval. We provide a detailed explanation of this produce in Appendix F.2. The high ESS values of MGR together with its accurate recovery of dynamical quantities provide strong evidence of the model’s reliability and accuracy.
> >
> > &nbsp;
> >
> > ### W4.
> > >The use of the strong law of large numbers in appendix C is not justified by any independence assumptions. For long trajectories, are there any mixing assumptions on the original and/or the perturbed process? Are the multiple trajectories at least independent?
> >
> > ### Response:
> >
> > We thank the reviewer for highlighting the issue of the Strong Law of Large Numbers (SLLN) in our Appendix C.
> >
> > **Regarding independence:**
> >  The Strong Law of Large Numbers (SLLN) for Markov chains used in Appendix C does not require independence in the i.i.d sense. Our justification follows the theory for ergodic Markov chains [1], which proves that for a Markov chain with a unique invariant distribution and suitable continuity of the transition operator, SLLN holds true. Hence, ergodicity and the existence of a stationary measure are sufficient for SLLN in Markov chains. In our work, we assume that the perturbed dynamics is ergodic, which is basic and fundamental to Markov process estimators.
> >
> > **Regarding multiple trajectories:**
> >  For ergodic SLLN for Markov chains, it is not required for multiple independent runs. However, multiple trajectories used in practice can effectively reduce finite-sample variance and increase effective sample size.
> >
> > In the revised manuscript, we have changed the cited reference and clarified the ergodic assumption.
> >
> > [1] Breiman, Leo. "The strong law of large numbers for a class of Markov chains." The Annals of Mathematical Statistics 31.3 (1960): 801-803.

---

> > > ### Author Response · Authors · 2025-11-20
> > >
> > > ### W5.
> > > > In line 128 it should be $t\in[T-\tau]$ instead; At the end of line 207 shouldn't it be $(x_t, x_{t+k\tau})$?; (4) is not really an identity, because (3) is not; Line 731: "We consider two probability on ..." delete $\mu$ and parentheses.
> > >
> > > ### Response:
> > >
> > > We thank the reviewer for the careful reading. We have corrected all the notational issues.
> > >
> > > >Eq. (4) is not really an identity, because Eq. (3) is not
> > >
> > > We would like to clarify that Eq. (3) is indeed an identity. It expresses the marginal transition ratio as the expectation of the path-wise Girsanov weight given the endpoints. Therefore, Eq. (4) also holds exactly at the theoretical level.
> > >
> > > While this equality is exact, it typically lacks a closed-form analytical solution because the conditional expectation over all path segments is intractable. Our proposed method in Section 4 uses a machine-learning estimator to approximate this conditional expectation, thereby providing a practical way to learn the marginal transition ratio from data. Its consistency can be guaranteed under the assumptions that the previous iteration $w_{(k-1)\tau}$ is accurately estimated and that the dataset size is sufficiently large (as clarified in Section 4.2).
> > >
> > > Thanks again for your suggestions, which improve the clarity and precision of the manuscript.

---

### Official Review · Reviewer_Y4MV · 2025-10-26

**Soundness:** 3
**Presentation:** 4
**Contribution:** 3
**Rating:** 8
**Confidence:** 4

**Summary:**

The paper presents a machine learning method for performing marginal Girsanov Reweighting. The idea is simple and well-executed, demonstrating benefits on a suite of benchmarks. While simple, the proposed method could substantially improve rare event sampling in molecular dynamics (MD) and provide an alternative method for performing Bayesian posterior inference for stochastic differential equations (SDE) such as the Lotka-Volterra model. I have minor critiques and questions.

**Strengths:**

The authors clearly explain the method and relevant background, such as MD. The exposition of Girsanov reweighing is clear and discussion of related work is clear, and, to my knowledge, comprehensive of relevant papers. The paper's novelty is conceptual clarity and practical stabilization.
- Their SDE posterior point estimate could be used in Simulation-based inference (SBI) and we recommend the authors consider applications there.

**Weaknesses:**

The authors address many of the limitations of the paper in the choice of the neural network used as the classifier. I address a few minor critiques below.
- Not really Bayesian inference of the SDE if you're estimating a point value in the SDE problems
- Nit-pick importance sampled expectation is missing marginal likelihood in expansion of the posterior but that cancels with the introduction of reference distribution.
- Saying hypothesis in line 410 is a little confusing as I was ready for a hypothesis test. Threw me off for a second thinking what you meant and might be more straightforward to say "ground-truth" or "reference true parameter" instead.
- Related to the SBI note, while outside the scope of this paper, calibration of parameter estimates of SDEs in Bayesian posterior estimation is an important consideration and is missing here. Examining the performance of the posterior parameter estimate on a held-out true point estimate is a good start but often in real-world SDEs there's a requirement for calibration since ground truth is never really known. The method actually provides a classifier in the vein of classifier-based SBI that could adapt already-existing calibration checks for SBI classifiers; but, this is a nice-to-have.
- The authors mention TRAM as an alternative method for transition-based reweighting along with GR, so why wasn't that evaluated? This seems Important to compare.

**Questions:**

- The paper relies on sampling endpoint pairs $(x_t, x_{t+k\tau})$ from trajectories under the perturbed dynamics $\tilde{\mu}$ and uses the Strong Law of Large Numbers (SLLN) to justify convergence of weighted averages. How robust is MGR when the perturbed dynamics are not ergodic, or when the sampled trajectories do not adequately explore the full state space?

---

> ### Author Response · Authors · 2025-11-20
>
> We would like to express our appreciation for the reviewer’s detailed review. According to the comments, we have revised the manuscript to clarify the relationship between MGR and simualtion-based Bayesian inference methods, TRAM. We have also corrected some notational omissions. Thanks again for your constructive feedback.
>
> ### S1
> > Their SDE posterior point estimate could be used in Simulation-based inference (SBI) and we recommend the authors consider applications there.
>
> ### Response:
>
> We thank the reviewer for pointing out the connection to simulation-based inference (SBI). We have added a dedicated discussion of SBI methods in the Related Work section (highlighted in blue in the revised manuscript), explicitly clarifying both the relationship and the key differences.
>
> We would like to emphasize an important distinction in problem setting. In classical SBI, it is assumed that simulation data under many different parameter values are available, and the SDE is treated as a black box. In contrast, the setting we study in paper is fundamentally different:
> - Only simulation data under a single reference parameter are available. For real molecular and physical processes, the reference parameter corresponds to the standard or nominal condition.
> - The physical effect of parameter perturbations on the drift term is known.
>
> SBI methods cannot be applied effectively in this single-reference setting because only one parameter realization is available. In contrast, MGR leverages reference-condition trajectories to compute transition likelihood ratios across parameters, which enables Bayesian parameter estimation even under sparsely sampled observations.
>
> We agree that integrating the strengths of MGR and SBI is an exciting future direction. Combining physical information with multi-parameter SBI techniques may further improve Bayesian inference for SDEs. We have added this point to the conclusion.
>
> &nbsp;
>
> ### W1.
> >Not really Bayesian inference of the SDE if you're estimating a point value in the SDE problems.
>
> ### Response:
>
> We thank the reviewer for raising this point. We are unsure what does "estimating a point value" mean here, and hope our reply addresses the reviewer’s question.
>
> We clarify that our goal in the Bayesian-inference experiments is not to estimate a single point value, but to recover the full posterior distribution $p(\theta\mid \mathbf y)$ for the SDE parameters with sparse observation $\mathbf y$.
>
> Our workflow follows the standard formulation of classical Bayesian inference for parametric SDEs:
> - the drift and diffusion structures are given analytically, but contain unknown parameters;
> - latent trajectories are simulated under a reference parameter $\theta_0$;
> - posterior distributions of parameters $\theta$ are obtained by evaluating the likelihood ratio relative to $\theta_0$ via MGR.
>
> We would be glad to elaborate further if any aspect has been misunderstood.
>
>
> &nbsp;
>
> ### W2.
> > Nit-pick importance sampled expectation is missing marginal likelihood in expansion of the posterior but that cancels with the introduction of reference distribution.
>
> ### Response:
>
> We thank the reviewer for pointing out this notational omission. In Eq. (S.3), the full expansion of the posterior should indeed include the marginal likelihood $p(\mathbf{y})$ in the denominator,
>
> $p(\theta \mid \mathbf{y})=\frac{p(\mathbf{y} \mid \theta) p(\theta)}{p(\mathbf{y})} .$
>
> We have corrected this expression in the revised supplementary material marked as "blue".
>
> As the reviewer correctly noted, once the posterior expectation is written in its importance-sampling form, the marginal likelihood will cancel automatically. The omission did not affect any of the derivations or experiments. We appreciate the reviewer’s attention to detail and have updated the equation.
>
> We hope this clarification addresses the reviewer’s question, and we would be glad to elaborate further if any aspect has been misunderstood.
>
>
> &nbsp;
>
> ### W3.
> > Saying hypothesis in line 410 is a little confusing as I was ready for a hypothesis test. Threw me off for a second thinking what you meant and might be more straightforward to say "ground-truth" or "reference true parameter" instead.
>
> ### Response:
>
> We thank the reviewer for pointing this out. The term "hypothesis" in experiments was indeed misleading, as our experiments do not involve any hypothesis testing. What we intended to denote is simply the reference (ground-truth) parameter used to generate the synthetic data. We have revised the manuscript accordingly by replacing "hypothesis" with "reference parameter", marked in "blue". We appreciate the reviewer’s suggestion for improving clarity.

---

> > ### Author Response · Authors · 2025-11-20
> >
> > ### W4.
> > > Related to the SBI note, while outside the scope of this paper, calibration of parameter estimates of SDEs in Bayesian posterior estimation is an important consideration and is missing here. Examining the performance of the posterior parameter estimate on a held-out true point estimate is a good start but often in real-world SDEs there's a requirement for calibration since ground truth is never really known. The method actually provides a classifier in the vein of classifier-based SBI that could adapt already-existing calibration checks for SBI classifiers; but, this is a nice-to-have.
> >
> > ### Response:
> >
> > We thank the reviewer for this thoughtful remark. We are unsure whether we have understood the calibration correctly, and we hope that our reply addresses the reviewer’s question. We would be glad to provide further clarification if any part of our response reflects a misunderstanding.
> >
> > Calibration is an important consideration in Bayesian inference when the ground-truth parameters are unknown. In the classical Bayesian SDE setting considered in this paper, calibration is typically assessed through the recovery of posterior distributions on sparse observations, since real-world applications rarely provide access to true parameter values. Our experiments already demonstrate that MGR successfully reconstructs the full posterior, which serves as the standard validation for this regime.
> >
> > More advanced calibration diagnostics---such as simulation-based calibration (SBC) or classifier-based calibration checks used in SBI---require repeated posterior inference over many synthetic datasets or amortized training over large simulation. Because MGR is designed for the non-amortized, physics-informed SDE setting, where inference is performed around a single reference process rather than across a large parameter space, these SBI-style calibration tools are not directly applicable.
> >
> > Nevertheless, we agree that adapting MGR to support hybrid physics-informed SBI-style calibration checks is an interesting direction for future research.
> >
> > &nbsp;
> >
> > ### W5.
> > > The authors mention TRAM as an alternative method for transition-based reweighting along with GR, so why wasn't that evaluated? This seems Important to compare.
> >
> > ### Response:
> >
> > We thank the reviewer for asking about TRAM and its relation to our work. TRAM and related methods (dTRAM, xTRAM) play an important role in combining biased and unbiased simulations. However, they solve a different class of problems from the one addressed by MGR, and are therefore not directly comparable as baselines. We have clarified this distinction in the revised Section 2 (Related Work), highlighted in "blue".
> >
> > **Features of TRAM**
> >
> > TRAM is formulated in the setting of multi-ensemble equilibrium sampling, where one collects equilibrium trajectories from several thermodynamic states (e.g., different temperatures or bias potentials) and constructs a joint Markov state model (MSM). Its main advantage is that it can directly estimate the equilibrium distribution for new, unseen biased dynamics, although it cannot directly recover dynamic information. Its theoretical foundation relies on:
> > - availability of multiple ensembles with known biases;
> > - simulation data satisfying local/global equilibrium and detailed balance;
> > - the validity of a MSM that couples thermodynamic and kinetic quantities.
> >
> > In our setting, TRAM cannot construct a unbiased MSM with simulations from a single specific accelerated simulation dynamic. Moreover, TRAM can not provide an explicit path-wise reweighting factor between transition pairs.
> >
> > **Features of MGR**
> >
> > In contrast, MGR addresses a different problem. It operates in the setting where the drift bias of the simulation dynamics is known, and the goal is to explicitly estimate the reweighting ratio of two transition dynamics. MGR computes transition ratios by learning a variance-reduced estimator of the Girsanov path-weight, which does not rely on any MSM construction, nor does it assume equilibrium. MGR can explicitly provide a reweighting ratio for given transition pairs, which is TRAM cannot achieve directly.
> >
> > Therefore, although both TRAM and MGR provide an unbiased estimation, they operate under different assumptions, target different mathematical objects, and yield different types of dynamical information. For this reason, a direct numerical comparison would not be meaningful. However, MGR can serve as a complementary component to TRAM methods, which falls outside the scope of the present work. Thanks again for your suggestion.

---

> > > ### Author Response · Authors · 2025-11-20
> > >
> > > ### Q1.
> > > > The paper relies on sampling endpoint pairs $(x_t, x_{t+k\tau})$ from trajectories under the perturbed dynamics $\tilde \mu$ and uses the Strong Law of Large Numbers (SLLN) to justify convergence of weighted averages. How robust is MGR when the perturbed dynamics are not ergodic, or when the sampled trajectories do not adequately explore the full state space?
> > >
> > > ### Response:
> > >
> > > We thank the reviewer for this question. Our use of the Strong Law of Large Numbers (SLLN) for Markov chains follows the standard assumption in reweighting and importance sampling: the perturbed dynamics is assumed to be ergodic with respect to its invariant measure, and the observable of interest is integrable. This assumption is fundamental to Markov-process–based estimators and is shared by all transition-based reweighting approaches. We have added this assumption in the revised Section 4.1 and Appendix C, marked in blue.
> > >
> > > If the perturbed dynamics is not ergodic, the trajectory remains confined to a single ergodic component. When the sampled trajectories explore only a strict subset of the state space, then no reweighting method---MGR, Girsanov Reweighting, TRAM, or otherwise---can recover expectations in regions that are never visited. In that case, the estimates are only valid on the ergodic component actually explored. This limitation is fundamental but not specific to MGR: reweighting necessarily requires that the support of the target dynamics is contained in the support of the sampling dynamics.
> > >
> > > We further emphasize that, unlike methods that assume global or local equilibrium (e.g., TRAM), MGR does not require the perturbed dynamics to be at equilibrium, which is also one of its key advantages. It applies equally to non-equilibrium enhanced-sampling trajectories, where obtaining well-equilibrated data can be difficult or computationally prohibitive.
> > >
> > > &nbsp;
> > >
> > > We are truly grateful for the remarks, which have been extremely helpful in improving the rigor and quality of our manuscript.

---

### Official Review · Reviewer_o39R · 2025-10-31

**Soundness:** 3
**Presentation:** 3
**Contribution:** 2
**Rating:** 4
**Confidence:** 2

**Summary:**

This paper introduces **Marginal Girsanov Reweighting (MGR)**, a method that combines classical stochastic reweighting theory with modern neural ratio estimation.   The classical **Girsanov reweighting (GR)** technique allows expectations under one stochastic differential equation (SDE) to be recovered from simulations under a different, typically more tractable SDE. However, as also shown by the authors GR suffers from **exponential variance growth** with time horizon, making it unstable for long trajectories. MGR mitigates this by **marginalizing over intermediate paths**: instead of pathwise weights, it learns **endpoint-based density ratios** using a **classifier-based neural ratio estimator**. The method bootstraps short-lag GR weights (where the variance is still manageable) to compose longer-lag ratios iteratively, yielding stable, scalable weights. Experiments on molecular dynamics (umbrella-sampled four-well and alanine dipeptide systems) and Bayesian SDE inference (Ornstein–Uhlenbeck and Lotka–Volterra models) show that MGR produces accurate long-horizon estimates where GR fails.

**Strengths:**

- **Well written and methodologically sound**: The paper presents a technically solid and clearly explained method, bridging stochastic analysis and modern density ratio estimation. The theoretical results (variance analysis, consistency proofs, and training algorithm) are rigorous.
- **Comprehensive empirical evaluation** The two application domains—molecular dynamics and Bayesian SDE inference—are complementary and well chosen. Results are convincing: MGR maintains higher effective sample size (ESS), better implied timescales, and accurate posterior estimates, even when GR and standard inference methods fail.

**Weaknesses:**

- **Related work:** The approach (specifically in the context of Bayesian inference for SDEs) is very much related to simulation-based inference [1] methods (NRE, NLE and specific specialization of these for SDEs). These should be atleast discussed in related work. Specifically [2,3] also a targeted towards Bayesian inference on SDEs and also are motivated by learning a certain property of transition densities that can be used to perform inference.
[1] Cranmer, Kyle, Johann Brehmer, and Gilles Louppe. "The frontier of simulation-based inference." _Proceedings of the National Academy of Sciences_ 117.48 (2020): 30055-30062.
[2] Gloeckler, Manuel, et al. "Compositional simulation-based inference for time series." _arXiv preprint arXiv:2411.02728_ (2024).
[3] Cai, Xin, et al. "Simulation-based transition density approximation for the inference of SDE models." _arXiv preprint arXiv:2401.02529_ (2023).

- Motivation can be made clearer:  The overall motivation can be made more clear for readers not familiar with GR or molecular dynamics some parts are somewhat confusing i.e. "direct simulation from the target law is often infeasible ..., which motivates us to introduce a perturbation ... perturbed dynamics are  easier to simulate ..." (page 3, top). But then GR reweighting requires anyway to evaluate both drifts $f$ and $\tilde{f}$  along the trajectory which has the same cost than the unperturbed process... . The answer, is somwhat implicit in the text, should be stated explicitly: *direct simulation of the unperturbed SDE may be computationally infeasible due to slow mixing or interest in rare events (under the target law)*. GR then allows to bias towards e.g.  rare events and correct it later.

**Questions:**

- For the Bayesian inference experiments, a _reference process_ must be defined, i.e., a particular choice of parameters and initial conditions under which the trajectories are simulated. While the manuscript specifies which reference settings were used, it does not explain **why** these choices were made or how they affect performance. Since the stability and effectiveness of MGR likely depend on how close the reference dynamics are to the true (target) parameters, a discussion of **how to select or tune the reference process in practice** would be very valuable. In its current form, it is unclear how one would make this choice from scratch, or how sensitive the method is to this design decision.

---

> ### Author Response · Authors · 2025-11-20
>
> We thank the reviewer for the positive and constructive feedback. The questions raised helped us clarify the relationship between MGR and simulation-based Bayesian inference methods, which we have now discussed in the revised manuscript. We address concerns point by point below.
>
> ### W1.
> > Related work: The approach (specifically in the context of Bayesian inference for SDEs) is very much related to simulation-based inference [1] methods (NRE, NLE and specific specialization of these for SDEs). These should be atleast discussed in related work. Specifically [2,3] also a targeted towards Bayesian inference on SDEs and also are motivated by learning a certain property of transition densities that can be used to perform inference.  <br>[1] Cranmer, Kyle, Johann Brehmer, and Gilles Louppe. "The frontier of simulation-based inference." Proceedings of the National Academy of Sciences 117.48 (2020): 30055-30062. <br>[2] Gloeckler, Manuel, et al. "Compositional simulation-based inference for time series." arXiv preprint arXiv:2411.02728 (2024).  <br>[3] Cai, Xin, et al. "Simulation-based transition density approximation for the inference of SDE models." arXiv preprint arXiv:2401.02529 (2023).
>
>
> ### Response:
>
> We thank the reviewer for pointing out the connection to simulation-based inference (SBI) and for listing relevant references. Following the suggestion, we have added a dedicated discussion of SBI methods in the Related Work section (highlighted in blue in the revised manuscript), explicitly clarifying both the relationship and the key differences.
>
> We would like to emphasize an important distinction in problem setting. In classical SBI, it is assumed that simulation data under many different parameter values are available, and the SDE is treated as a black box. In contrast, the setting we study in the paper is fundamentally different:
> - Only simulation data under a **single** reference parameter are available. For real molecular and physical processes, the reference parameter corresponds to the standard or nominal condition.
> - The physical effect of parameter perturbations on the drift term is known.
>
> SBI methods cannot be applied effectively in this single-reference setting because only one parameter realization is available. In contrast, MGR leverages reference-condition trajectories and physical effect to compute transition likelihood ratios across parameters, which enables Bayesian parameter estimation even under sparsely sampled observations.
>
> We agree that integrating the strengths of MGR and SBI is an exciting future direction. Combining physical information with multi-parameter SBI techniques may further improve Bayesian inference for SDEs. We have added this point to the conclusion.

---

> > ### Author Response · Authors · 2025-11-20
> >
> > ### W2.
> > > Motivation can be made clearer: The overall motivation can be made more clear for readers not familiar with GR or molecular dynamics some parts are somewhat confusing i.e. "direct simulation from the target law is often infeasible ..., which motivates us to introduce a perturbation ... perturbed dynamics are easier to simulate ..." (page 3, top). But then GR reweighting requires anyway to evaluate both drifts $f$ and $\tilde f$ along the trajectory which has the same cost than the unperturbed process... . The answer, is somwhat implicit in the text, should be stated explicitly: direct simulation of the unperturbed SDE may be computationally infeasible due to slow mixing or interest in rare events (under the target law). GR then allows to bias towards e.g. rare events and correct it later.
> >
> > ### Response:
> >
> > We thank the reviewer for pointing out the ambiguity in the motivation and have revised the corresponding paragraph (marked in "blue" in the Sections 1, 2 & 3) accordingly.
> >
> > **Motivation clarification**
> >
> > The original text combined two application regimes (MD kinetic reweighting and Bayesian SDE inference) together in Section 3.1 (page 3), which made the motivation insufficiently explicit. We now clearly state the key point: in molecular dynamics, direct simulation under the target drift $f$ may be computationally infeasible due to slow mixing or the rare events of interest under the target law. In such cases, simulating long, unbiased trajectories requires prohibitively long computation. A perturbed drift $\tilde f$ is designed to accelerate exploration, e.g., flattening barriers or biasing the system toward important but rare configurations. This makes the trajectories much easier to sample. Girsanov’s theorem then provides an exact likelihood ratio that corrects the bias and recovers expectations under the original dynamics.
> >
> > **Computational cost clarification**
> >
> > While GR requires evaluating $f-\tilde f$ along sampled trajectories, its cost depends heavily on the application.
> > In Bayesian SDE inference, both $f$ and $\tilde f$ should be evaluated to compute the ratio. However, in molecular dynamics (enhenced sampling), the term $f-\tilde f$ usually related to some low dimensional reaction coordinates rather than the full state space. For example, in alanine dipeptide, the bias term is only related to two important dihedral angles in order to accelerate simulation. As shown in [1], GR yields minimal computational overhead with substantial mixing acceleration.
> >
> >
> > [1] Donati, Lorenzo, Carsten Hartmann, and Bettina G. Keller. "Girsanov reweighting for path ensembles and Markov state models." The Journal of chemical physics 146.24 (2017).

---

> > > ### Author Response · Authors · 2025-11-20
> > >
> > > ### Q1.
> > > > For the Bayesian inference experiments, a reference process must be defined, i.e., a particular choice of parameters and initial conditions under which the trajectories are simulated. While the manuscript specifies which reference settings were used, it does not explain why these choices were made or how they affect performance. Since the stability and effectiveness of MGR likely depend on how close the reference dynamics are to the true (target) parameters, a discussion of how to select or tune the reference process in practice would be very valuable. In its current form, it is unclear how one would make this choice from scratch, or how sensitive the method is to this design decision.
> > >
> > > We thank the reviewer for raising this important point. We clarify how the reference process is chosen in practice and why MGR is not overly sensitive to this choice.
> > >
> > > **How to choose the reference process**
> > >
> > > Unlike amortized SBI methods that simulate trajectories over a wide parameter range, MGR relies on a single reference process to generate latent paths and sparse observations. In classical Bayesian SDE inference, a standard practice is to select a numerically stable, physically meaningful parameter value as the reference. For real molecular and physical processes,
> > > the reference parameter corresponds to the standard or nominal condition. Here, we adopt the same setting as in [1].
> > >
> > > **How to determine a practical range of candidate parameters**
> > >
> > > In practice, the candidate parameters are typically chosen as small perturbations around the reference value, since the reweighting estimator remains accurate only when the reference and target dynamics have sufficient overlap. But we can also start from a broader candidate range. We evaluate the ESS values of the short-lag Girsanov weights to assess stability: parameters leading to extremely low ESS indicate poor overlap and unreliable reweighting. We therefore retain only those parameter values whose ESS exceeds a minimal threshold. This procedure adaptively identifies the region in which the reference dynamics provide adequate coverage of the target dynamics, ensuring that MGR reweighting remains statistically reliable and well-conditioned. The corresponding analysis has beed added in the revised manuscript Appendix F.3 marked in "blue".
> > >
> > >
> > >
> > > [1] Ghosh, Sanmitra, Paul J. Birrell, and Daniela De Angelis. "Differentiable Bayesian inference of SDE parameters using a pathwise series expansion of Brownian motion." International Conference on Artificial Intelligence and Statistics. PMLR, 2022.
> > >
> > > &nbsp;
> > >
> > > Thanks again for the valuable feedback, which has strengthened the clarity and overall quality of our work.

---

### Official Review · Reviewer_Gjth · 2025-11-07

**Soundness:** 3
**Presentation:** 3
**Contribution:** 3
**Rating:** 4
**Confidence:** 2

**Summary:**

The paper proposes marginalised Girsanov weighting method, where the standard Girsanov weighting between trajectories is extended to between measures.

**Strengths:**

The method is principled and well motivated. The method is original, and seems to be effective at small demonstrations. The paper is overall clearly presented.

**Weaknesses:**

The work does not demonstrate much significance. The results are toy'ish small demonstrations, where the innovation does work over the vanilla method. There is however no "real-world" experiments, or any comparisons to competing methods other than the vanilla Girsanov.

The novelty is also somewhat limited as the work combines known techniques in the ratio estimation.

The work could also improve clarity by visualising the setting with some SDE paths or measures that highlight the problem of using drift f, and show the need to move towards tilde{f}

**Questions:**

See above

---

> ### Author Response · Authors · 2025-11-20
>
> We sincerely thank the reviewer for the comments. The feedback helps us better position our contribution within the landscape of recent literature. Based on your suggestion, we have added a visualization to show the effect of enhanced sampling and MGR.
>
> ### W1.
> > The work does not demonstrate much significance. The results are toy'ish small demonstrations, where the innovation does work over the vanilla method. There is however no "real-world" experiments, or any comparisons to competing methods other than the vanilla Girsanov.
>
> ### Response:
>
> We thank the reviewer for raising this concern. The reviewer’s main concern centers on the practical significance and experimental scope of our study. To address this concern comprehensively, we clarify below:
>
> **About the role of Girsanov Reweighting (GR)**
>
> GR is currently the most widely used and most established mathematically rigorous method for recovering unbiased dynamical properties from nonequilibrium simulation trajectories [1–2]. Prior work in this direction has almost exclusively focused on how to compute the path-space weights efficiently during the simulation (see, e.g., [3]). However, a fundamental theoretical limitation remains: the variance of Girsanov path weights grows exponentially with trajectory length (Sec. 3.2), which makes long-timescale dynamical analysis intractable. This limitation is not due to computation, but due to the variance structure of GR itself (Appendix A).
>
> **About the contribution of Marginal Girsanov Reweighting (MGR)**
>
> To the best of our knowledge, MGR is the first method that directly overcomes this variance barrier in a theoretically principled way. The key observation is that many downstream dynamical analyses depend not on the full path, but on the transition probability between two states at a chosen lag time. Based on this insight, MGR uses density-ratio estimation (DRE) to compute the average contribution of intermediate states (Eq. (3)), which dramatically reduces variance and extends the applicable time horizon of GR. This represents a conceptual shift **from path-space weighting to marginal transition weighting**, rather than an incremental improvement.
>
> We will clarify this conceptual distinction and the methodological significance more explicitly in the revised version.
>
> **About the experimental systems**
>
> Regarding the concern about "toy-ish" experiments and "real-world" relevance, we would like to stress that the systems we use are standard benchmarks in their respective communities. In molecular simulation, alanine dipeptide is a standard benchmark in MD methods development as it exhibits all the hallmark properties of larger systems, including meta-stability and slow mixing, while being tractable through brute force simulation. Likewise, the one-dimensional four-well potential captures metastable kinetics in a controlled setting, while the Ornstein–Uhlenbeck and Lotka–Volterra processes are classical models in Bayesian inference and stochastic dynamics. These systems provide reliable ground truth and allow us to carefully diagnose variance, stability, and bias.
>
> While we agree that applying MGR to larger systems (e.g., multi-window umbrella sampling for larger proteins) is important we reserve this next step for future work, and we view the present work as establishing the methodological foundation.
>
> [1] Donati, Lorenzo, Carsten Hartmann, and Bettina G. Keller. "Girsanov reweighting for path ensembles and Markov state models." The Journal of chemical physics 146.24 (2017).
>
> [2] Kieninger, Stefanie, Simon Ghysbrecht, and Bettina G. Keller. "Girsanov reweighting for simulations of underdamped Langevin dynamics. Theory." arXiv preprint arXiv:2303.14696 (2023).
>
> [3] Schafer, Joana-Lysiane, and Bettina G. Keller. "Implementation of Girsanov Reweighting in OpenMM and Deeptime." The Journal of Physical Chemistry B 128.25 (2024): 6014-6027.

---

> > ### Author Response · Authors · 2025-11-20
> >
> > ### W2.
> > >The novelty is also somewhat limited as the work combines known techniques in the ratio estimation.
> >
> > ### Response:
> >
> > We thank the reviewer for the comment. We would like to clarify that the novelty of our work does **not** reside in the straightforward application or combination of existing ratio–estimation techniques. Instead, it lies in the **conceptual and theoretical** development of the *marginal Girsanov reweighting* procedure. This method is fundamentally new and enables transition-based reweighting that yields stable and efficient long-lag Girsanov weights. Addressing long-lag transition reweighting has been a longstanding challenge in MD, and to the best of our knowledge, no existing approaches explicitly tackle this problem by directly estimating the ratio.
> >
> > Our key idea is to replace the unstable Girsanov weight with a marginalized and iteratively training likelihood-ratio estimator. This strategy is unique to our setting and, to the best of our knowledge, has not appeared in prior work.
> >
> > We selected a classifier-based ratio estimator because it aligns well with the structure of our algorithm and yields stable estimates in our experiments. As reported in Appendix F.1, we experimented with several state-of-the-art ratio estimation techniques, and none provided substantial improvements. We will continue to explore advanced density-ratio estimators to further enhance the model's capability.
> >
> > &nbsp;
> >
> > ### W3.
> > >The work could also improve clarity by visualising the setting with some SDE paths or measures that highlight the problem of using drift f, and show the need to move towards $\tilde{f}$.
> >
> > ### Response:
> >
> > We appreciate the reviewer’s suggestion. Visualizing the dynamics under the original drift versus the perturbed drift is useful for understanding the necessity of accelerated simulation.
> >
> > The difficulty in traditional MD is that, under the unbiased drift $f$, the SDE often exhibits slow mixing, rare barrier-crossing events, and highly imbalanced state occupancy. In such cases, unbiased trajectories do not provide enough transitions to estimate kinetic quantities at the desired lag times. This motivates the use of a modified drift $\tilde{f}$ that lowers barriers and produces trajectories with significantly richer transition statistics. This strategy is commonly referred to as ``enhanced sampling'' in the MD community.
> >
> > To make this contrast explicit, we now highlight the following visualizations:
> >
> > - Figure 1 (double-well) shows how altering the drift effectively lowers the potential barrier, leading to visibly more frequent well-to-well transitions.
> >
> > - Figure S.1 (four-Well) includes energy plots of both unbiased and biased dynamics. The biased drift introduces a repulsive region near the center, which effectively increases transitions between the two side wells. Eigenfunctions in Figure 3 also clearly demonstrates this point.
> >
> > - We have also added a density plot of the transfer operator in Figure S.4 to visualize the difference of transition probability. (Marked as "blue" in the revised manuscript in Page 18-19.)
> >
> > - Figure S.5 (alanine dipeptide) visualizes the energy function after biasing along dihedral angles, which can promotes transitions.
> >
> >
> > We would like to once again thanks for the valuable feedback.

---

### Author Response · Authors · 2025-12-04
**Final Remarks**

We thank the AC for their efforts in overseeing our submission, particularly given the increased workload of this year’s review cycle. We are also deeply grateful to the reviewers for the time and care, which have enhanced the quality of the manuscript.

Our work focuses on recovering unbiased dynamical properties from biased/perturbed simulations. We address the variance-explosion challenge in classical Girsanov reweighting by marginalizing over intermediate states. Using neural ratio estimation, our work can construct stable and scalable weights for long-timescale processes.

During the rebuttal, we addressed all reviewer comments and added several targeted experiments, with all updates highlighted in **blue** in the revised draft. Unfortunately, we did not receive any responses before the discussion period closed. Below, we summarize the key points of the discussion.

- Reviewer o39R and Reviewer Y4MV inquired about how our method relates to simulation-based inference (SBI). Accordingly, we expanded the **Bayesian inference for SDEs** part of Section 2 to emphasize the fundamental difference: whereas SBI treats the SDE as a black box and relies on simulations generated under many different parameter settings, MGR leverages explicit knowledge of how parameter perturbations affect the drift term and requires simulation data only under a **single** reference parameter. Therefore, a direct comparison or application is not available.

- Reviewer Ji2i pointed out an additional baseline [1] that could serve as a comparison. We conducted the corresponding experiments and confirmed that our model outperforms this baseline (https://figshare.com/s/0ce048e7e9cec6d01ac7). Moreover, our method offers a more broadly applicable paradigm:
  - The baseline relies on the Koopman operator and directly estimates the reweighted eigenfunction, whereas our approach explicitly learns the reweighting ratio, enabling downstream applications such as Bayesian inference.
  - The baseline requires simulations to reach equilibrium, while our method does not impose this constraint.

- Reviewer Gjth raised concerns regarding the significance and novelty of our work. To the best of our knowledge, MGR is the first method that overcome the variance barrier of Girsanov reweighting in a theoretically principled way. Addressing long-lag transition reweighting has been a longstanding challenge in MD, and our work explicitly tackle this problem by directly estimating the ratio.

We sincerely thank the reviewers for their thoughtful feedback, which has improved both the clarity and rigor of the manuscript. We remain confident that MGR offers a principled and effective solution to the long-standing variance challenge in Girsanov reweighting. We also thank the AC for the careful handling of our submission.

[1] Devergne T, Kostic V, Parrinello M, et al. From biased to unbiased dynamics: An infinitesimal generator approach[J]. Advances in Neural Information Processing Systems, 2024, 37: 75495-75521.

---

### Meta-Review · Area_Chair_ECKU · 2026-01-07

**Summary:**

The paper introduces Marginal Girsanov Reweighting (MGR), a method designed to mitigate the variance explosion inherent in classical Girsanov reweighting (GR) for long-horizon simulations. By marginalizing over intermediate paths using neural ratio estimation, the authors provide a stable way to recover unbiased properties from biased or perturbed simulations, with applications in molecular dynamics and Bayesian parameter inference for SDEs. While the method addresses a relevant challenge in rare-event sampling, the overall consensus after review and rebuttal leans toward the work requiring further maturation.

**Reviewer Concerns:**

The authors clarified the relationship between MGR and Simulation-Based Inference (SBI), highlighting that MGR functions with a single reference parameter rather than multi-parameter simulations. A comparison with the infinitesimal generator baseline (Devergne et al., 2024) was also provided, showing that MGR can handle underdamped Langevin dynamics where the baseline may struggle.

Reviewer Ji2i noted that the theoretical support remains weak, particularly regarding how noise from discretized Girsanov steps does not accumulate during the recursive training procedure. While experiments were conducted on standard benchmarks like alanine dipeptide, reviewers felt the demonstrations remained somewhat "toy-ish" and lacked validation on truly complex, real-world systems beyond these diagnostic cases. The transition between continuous and discretized processes was described as opaque, making the methodological foundation difficult to verify fully.

**Reviewer Scores:**

Reviewer Y4MV: Likely to stay at 8 (Accept). They valued the conceptual clarity but their specialized focus on MD did not fully account for the broader theoretical skepticism.

Reviewer Ji2i: Likely to stay at 4 (Reject). Despite the rebuttal, the core concerns regarding theoretical noise accumulation and the recursive training's stability were not resolved to an expert level.

Reviewer o39R: Likely to stay at 4 (Reject). They maintained that the novelty was somewhat limited by the reliance on existing ratio estimation techniques.

Reviewer Gjth: Dismissed in final evaluation due to brevity, but originally scored a 4 (Reject) citing limited significance.

---

### Decision · Program_Chairs · 2026-01-26

Reject